# Simultaneous selection of nanobodies for accessible epitopes on immune cells in the tumor microenvironment

Thillai V. Sekar [1,2,6], Eslam A. Elghonaimy [1,6], Katy L. Swancutt [1],
Sebastian Diegeler [1], Isaac Gonzalez [1], Cassandra Hamilton [1],
Peter Q. Leung [1], Jens Meiler[3,4], Cristina E. Martina [3], Michael Whitney[5] &
Todd A. Aguilera [1] ✉

In the rapidly advancing field of synthetic biology, there exists a critical need for technology to discover targeting moieties for therapeutic biologics. Here we present INSPIRE-seq, an approach that utilizes a nanobody library and next-generation sequencing to identify nanobodies selected for complex environments. INSPIRE-seq enables the parallel enrichment of immune cell-binding nanobodies that penetrate the tumor microenvironment. Clone enrichment and specificity vary across immune cell subtypes in the tumor, lymph node, and spleen. INSPIRE-seq identifies a dendritic cell binding clone that binds PHB2. Single-cell RNA sequencing reveals a connection with cDC1s, and immunofluorescence confirms nanobody-PHB2 colocalization along cell membranes. Structural modeling and docking studies assist binding predictions and will guide nanobody selection. In this work, we demonstrate that INSPIRE-seq offers an unbiased approach to examine complex microenvironments and assist in the development of nanobodies, which could serve as active drugs, modified to become drugs, or used as targeting moieties.

Cancer immunotherapy has changed the treatment landscape for some cancers, but variable responses reflect limitations imposed by the tumor microenvironment (TME)[1–3]. There is increasing recognition that the TME plays a critical role in tumor evolution and progression. Although tumor cell adaptations may be stochastic, we learned that modifications of known cancer drivers and targeting immunologic pathways can alter the TME quickly[4,5]. With new proteomic and genomic technologies, the TME can be interrogated in ways not previously attainable, which can nominate drug targets. For example, with mass cytometry we can understand the complex immune system networks and how they interact with various organs throughout the body and the response to cancer immunotherapy[6,7]. Single cell RNA sequencing (scRNAseq) has provided critical perspective on composition and immune cell interactions in the TME[8,9]. Aside from challenges in physical assessments, such approaches can descriptively and functionally assess the TME's immunologic nature and nominate potential targets. The challenge of target validation and developing therapies that can access nominated targets remain. Customized drug development pipelines must follow target validation.

Bacteriophage display technology revolutionized the development of biologic therapies for known targets with in vitro approaches[10]. In vivo selection with phage libraries first demonstrated organ targeting and identified peptides that could have therapeutic activity[11–13]. We later used this approach to select enzymatically-cleaved peptides in the tumor microenvironment (TME) and to identify nerve-binding peptides for use in fluorescence-guided surgery[14,15]. The

[1]Department of Radiation Oncology, the University of Texas Southwestern Medical Center, Dallas, TX, USA. [2]Department of Microbiology, Pondicherry University, Kalapet, Puducherry, India. [3]Department of Chemistry and Center for Structural Biology, Vanderbilt University, Nashville, TN, USA. [4]Institute for Drug Discovery, Leipzig University Medical School, Leipzig, Germany. [5]Department of Pharmacology, University of California San Diego, La Jolla, CA, USA. [6]These authors contributed equally: Thillai V. Sekar, Eslam A. Elghonaimy. ✉e-mail: todd.aguilera@utsouthwestern.edu

traditional approach to identify enriched peptides, antibodies, or proteins by phage display is to sequence clones identified by high dilution using Sanger sequencing. With next generation sequencing (NGS), analysis can be deeper, more informative, and reveal the potential of in vivo selection adapted to various settings. In vivo phage selection has several advantages over ex vivo methods including but not limited to: it ensures that targets are accessible after systemic administration, it is specific for target epitopes linked to cellular activities of different target cells and tissue types, and it avoids biases introduced by organ excision and tissue disruption that might occur during ex vivo practices.

In this work, we devised INSPIRE-seq, an In vivo **N**anobody **S**election **P**ipeline for **I**mmune **R**epertoire identification in complex **E**nvironments by **seq**uencing. It is an in vivo phage display biopanning strategy to take steps toward an unbiased selection of nanobodies that bind to unaltered specific cell populations, in this case, tumor-infiltrating leukocytes (TIL) in triple negative breast cancer. We demonstrate biological parallel selection of enriched nanobodies for multiple immune cell subsets in the TME simultaneously. We then identified a nanobody enriched from dendritic cells (DCs) that targets Prohibitin-2 (PHB2) in the tumor and draining lymph nodes. Thus, nanobodies can be enriched for target cells and the target antigen can be identified. Further investigation of identified targets can open new avenues for research or drug development. Such drug development can be simplified where the selected nanobody could be the drug, the targeting moiety, or be modified to develop a drug, such as an antibody-drug conjugate.

## Results

### Strategy to select immune cell binding nanobodies in the TME

A camelid based VHH nanobody bacteriophage library was derived from peripheral blood mononuclear cells of twenty non-immunized llamas. VHH nanobodies are ideal because of their small size, variable region from a single heavy chain, stability, modularity, low immunogenicity due to their homology to human immunoglobulin VH3, and 90 to 130 amino acid length, which provides spatial coverage of unique epitopes[16–19]. To validate the INSPIRE-seq approach, we performed four rounds of biopanning by intravenous administration of the nanobody library, then amplified the enriched library and performed NGS from the CD45[+] cells after each round (Fig. 1a). We sorted five different immune cell subtypes to perform parallel enrichment that would allow for the identification of nanobodies for each cell type creating a distribution of selectivity (Fig. 1b). We isolated both tumor and lymph nodes from the Py8119 and Py117 syngeneic breast tumors, which have different immune responsiveness to therapy, making them complementary for selection[20,21]. We next developed a computational pipeline for sequence alignment, nanobody sequence identification, assessment of clonal dynamics, and selection of nanobodies based upon selectivity (Fig. 1c)

### NGS assesses the library and nanobodies in tissue during biopanning

To validate in vivo selection, we first detected the full-length nanobody sequences with the key variable regions by NGS. The computational pipeline for sequence analysis identified unique phage clones and their relative abundance. With stringent methods we reconstructed the VHH sequences with 600 bp average PCR amplicon length using miSeq PE300. This provided a high throughput assessment that would be difficult to achieve with the use of Sanger sequencing or long-read sequencing. As expected, we observed greater variability in the CDR3 than the CDR1 and CDR2 regions (Fig. 2a). The length of the CDR3 region varied with a median of 18 amino acids (range 1–36 amino acids), and the distribution was consistent over all rounds of biopanning in the tumor and lymph node (Fig. 2b and Supplementary Fig. 1a). To highlight diversity in the initial library, we assembled 20,083 VHH

nanobodies from 405,069 reads and calculated an average of one read per VHH, thereby alleviating concerns regarding PCR amplification artifacts in the library prior to in vivo selection (Supplementary Table 1 and Supplementary Fig. 1b, c).

We hypothesized that INSPIRE-seq could evaluate the selection of a diverse nanobody library in a complex system that targets different immune cell subtypes in the TME. To test this, we injected a library of approximately $1.5 \times 10^{12}$ phages intravenously, then 2 h later dissociated the tumor and draining lymph node (dLN) tissue as we had previously done when selecting for enzymatic cleavage substrates[15]. CD45[+] cells were separated and bacteria were infected with phages from cell lysates. The sample was titered and single clones were subsequently sequenced using Sanger methods separately, then pools from each cell subtype were amplified and sequenced using miSeq PE300 with approximately 450,000 reads per sample (Fig. 1a, b). Three tumor-bearing mouse replicates with Py8119 and Py117 tumors were used for each round of biopanning. We included the dLNs as a second tissue abundant with leukocytes that could corroborate and complement the TME findings. The percent unique VHHs and distribution of unique CDR regions were verified for each round (Supplementary Fig. 1b, c).

### INSPIRE-seq results in biologic enrichment in the TME and lymph node

Parallel enrichment, or simultaneous enrichment of nanobodies with different selectivity for each immune cell subtype, could be achieved using the diverse CD45[+] population for each biopanning round. Before the first biopanning (BP1), we sequentially selected bulk CD45[+] cells by ficoll gradient separation, then removed dead cells via magnetic beads to focus our highly diverse library towards leukocytes (BP0). For BP1-4 we isolated five different cell subtypes using a magnetic bead sorting scheme. A total of $4 \times 10^6$–$6 \times 10^7$ phages were isolated from each of five different CD45[+] cell subtypes from each round of biopanning (Fig. 1b). We selected for CD11b[+] myeloid cells, CD11c[+] DCs from the CD11b[−] flow through, then T cells from the CD11b[−] and CD11c[−] flow through. For T cells, we selected CD8[+], CD4[+]CD25[+] and CD4[+]CD25[−] populations. Flow cytometry confirmed enrichment of each cell population sorted using magnetic beads (Supplementary Fig. 2).

After verifying that specific VHH clones could be identified by INSPIRE-seq and that cell subtypes could be sorted efficiently, we evaluated whether in vivo biopanning could enrich specific nanobodies that bind to targets in the TME. We assessed the number of VHHs and the diversity for each round of biopanning, tumor type, and tissue type. We combined all VHH clones from each cell subpopulation for the tumor and dLN to calculate the proportion of sequences each clone occupied for each round of biopanning. The proportion of occupied repertoire space was plotted for all clones based on the rank of abundance as the top clone proportion, or by how many counts were identified for each clone as the rare clone proportion (Fig. 2c). Briefly, the library did not identify duplicate sequences for the same clone, most sequences were identified ten or fewer times in BP0, and in BP1-4 there was enrichment with multiple highly abundant clones having over 100 copies (Fig. 2c). We also observed this pattern of enrichment in the dLN and cell subpopulation (Supplementary Fig. 3). The repertoire space of the top ranked clones in each biopanning was expanded through biopanning rounds, with high representation of the ten most abundant clones (Fig. 2c).

To gain deeper perspective on the in vivo biopanning, we assessed the enrichment of VHHs through diversity extrapolation, shown in Fig. 2d. This approach considers the effect of the differences in sequence depth and the number of VHHs retrieved from each biopanning. The resulting rarefaction curves showed an immediate diversity reduction after BP1, which is consistent with the representation in Fig. 2c for the tumor and lymph node (Fig. 2d and Supplementary Fig. 4a). Diversity reduction was confirmed by true diversity

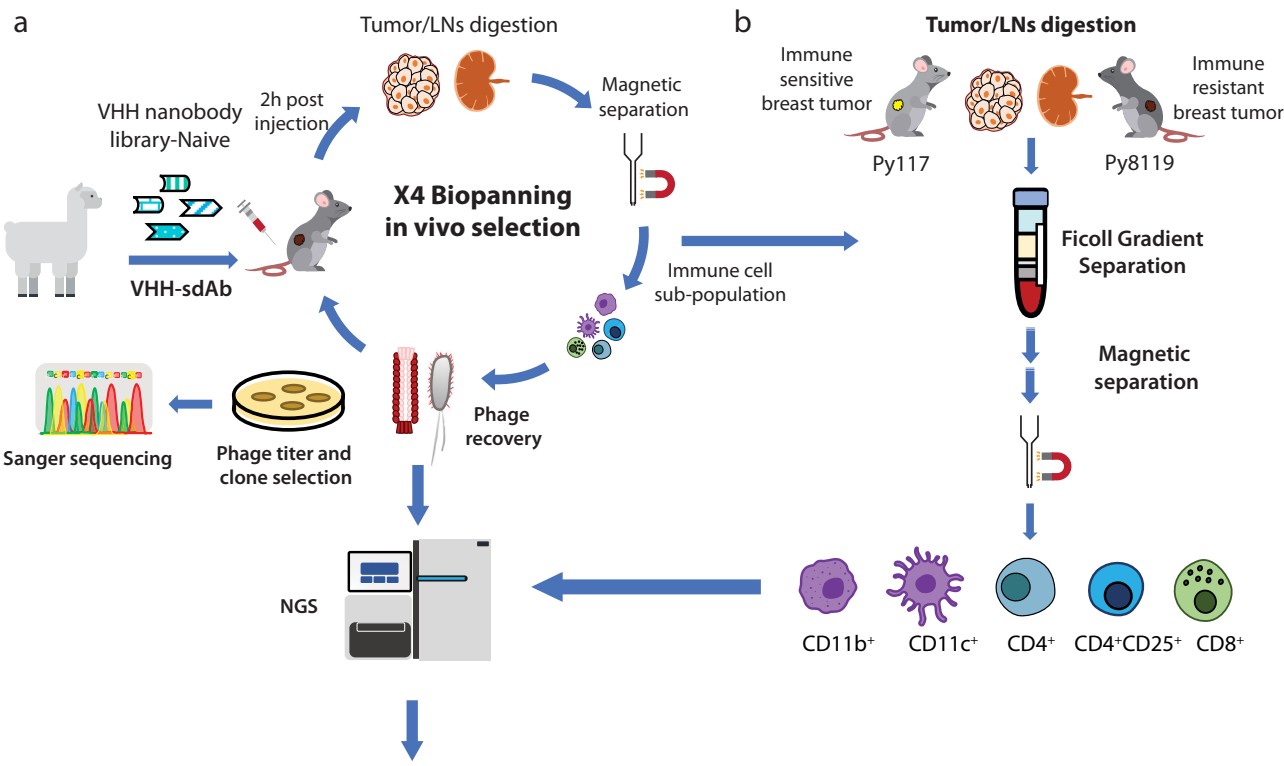

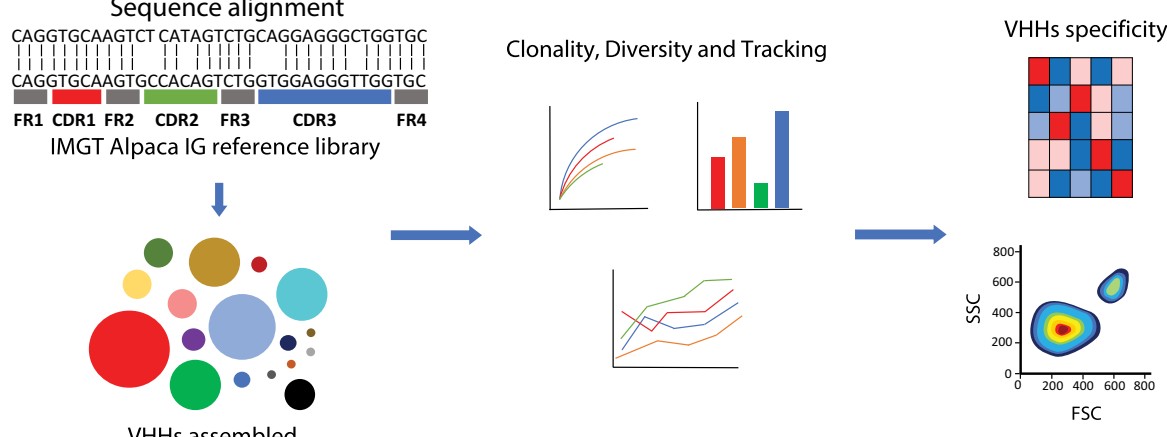

**Fig. 1 | In vivo biopanning for selection of VHH nanobodies that target the immune TME and assessment. a** In vivo biopanning schema of VHH-nanobody phage display harvesting tumor and draining LN analyzed by NGS and viral titer then amplified CD45⁺ cells for a total of four rounds. **b** Each immune cell sub-population was sorted from immune-sensitive (Py117) and immune resistant (Py8119) tumors and LNs via ficoll gradient followed by magnetic beads separation for each round, then samples were sequenced to evaluate selection and enrichment. **c** NGS analysis pipeline for each round of biopanning. Reads were aligned to Alpaca IG reference, all regions of VHHs CDRs were assembled, followed by clonality assessment, diversity assessment, and clone tracking for enrichment. Llama and mouse illustrations were adapted from https://www.svgrepo.com/svg/162/llama, https://creazilla.com/nodes/7772730-mouse-clipart under creative commons license (CC0).

estimations (Supplementary Fig. 4b). By focusing on the unique VHHs, we observed a reduction in the percentage of unique VHHs retrieved with each round of biopanning in the tumor (Fig. 2e) and dLN (Supplementary Fig. 1b).

**Parallel enrichment reveals nanobodies selective for cell subsets**

When taken together, it is clear that considerable enrichment occurred after BP1, which was consistent across different tumors, tissues, and cell sub-populations. This suggests that multiple rounds of biopanning are not needed to achieve a desirable amount of enrichment. We drew our attention to specific clones, which would further confirm biologic enrichment. The most abundant clones in BP0 disappeared during subsequent rounds of biopanning, showing that nonselective amplification of overrepresented clones did not occur (Fig. 3a, Supplementary Fig. 4 c, e, g). The opposite was true for clones that showed specific enrichment. The most abundant clones at the end of the biopanning had increased their fractional proportion of all sequences, supporting the idea that the greatest enrichment occurred between BP1 and BP2 (Fig. 3b, Supplementary

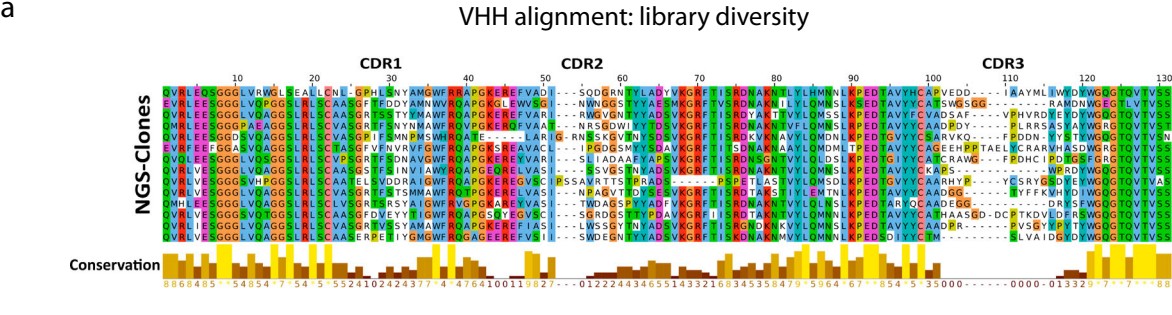

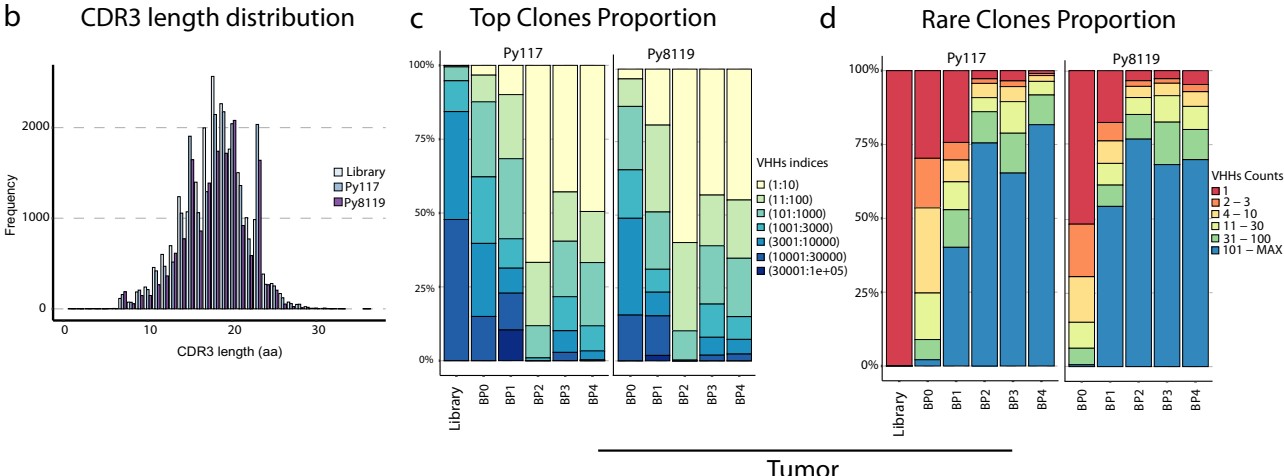

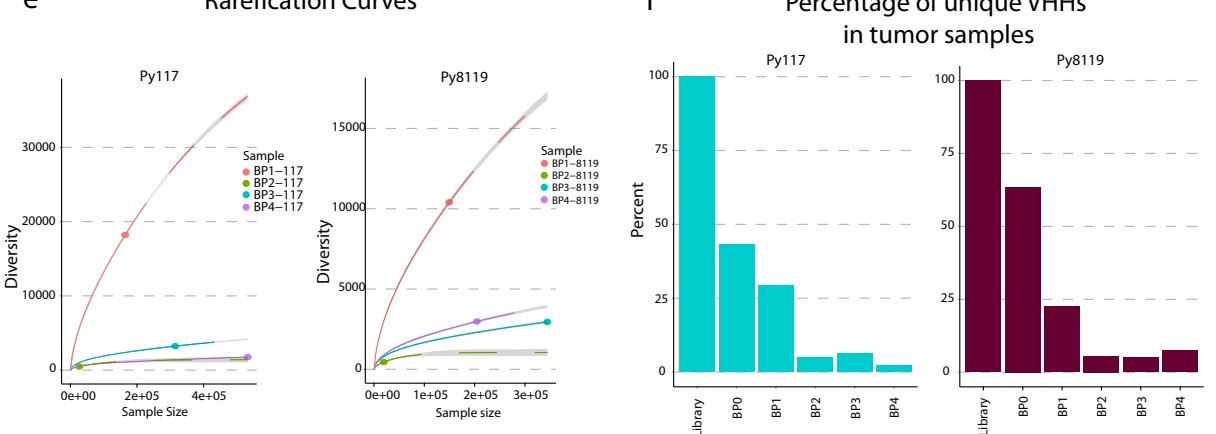

**Fig. 2 | Diversity and enrichment across in vivo biopanning to validate the approach. a** VHH alignment showing the diversity in CDR1, CDR2 and CDR3 regions. **b** Frequencies of CDR3 amino acid length in the library, Py117 and Py8119 samples demonstrate an equal distribution across the samples with a median length of 18 aa. Top clone proportion (**c**) and rare clonal proportion (**d**) showing summary proportion of VHHs with specific indices and counts in Py117 and Py8119 tumor samples from BP0 to BP4. **e** Rarefaction curve assessed the diversity of BP1 to BP4 in Py117 and Py8119 tumor samples through extrapolation and sub-sampling showing the reduction in diversity after BP1. **f** Percentage of unique VHH numbers in tumor samples of Py117 and Py8119.

Fig. 4d, f, h). This confirms that enrichment is achieved through a biological selection process.

The traditional approach to identifying enriched peptides, antibodies, or proteins by phage display is to sequence clones identified at a high dilution via Sanger sequencing. This captures the most abundant clones, such as a selected upper proportion depicted in Fig. 3c. We hypothesized that NGS would capture a broader distribution of clones that have biologic meaning and that a selective nanobody of relevance may not have the highest enrichment. To confirm that NGS

outperforms Sanger sequencing and accurately identifies enriched sequences, we performed Sanger sequencing on multiple clones from high dilution of BP3 and BP4 samples. There was a total 293 clones isolated from high dilution titer plates, and of these, 156 unique CDR3 regions demonstrated enrichment at the CDR3 level. Overall, 81.6% of all VHH clones were identified by NGS from BP3 or BP4, confirming that enriched clones could be identified by both methods (Supplementary Fig. 5a–f). A histogram plotting the Log2 frequency of counts on the x-axis by the proportional density of clones at that frequency on

a

### Most abundant clones from BP0 Py117

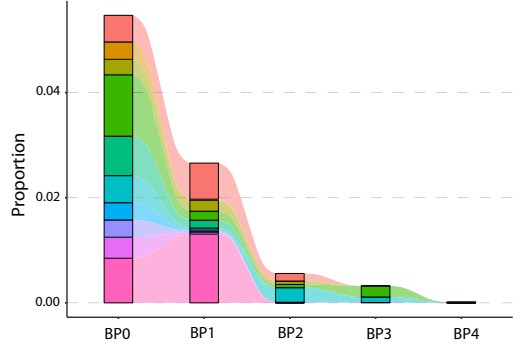

### Most abundant clones from BP0 Py8119

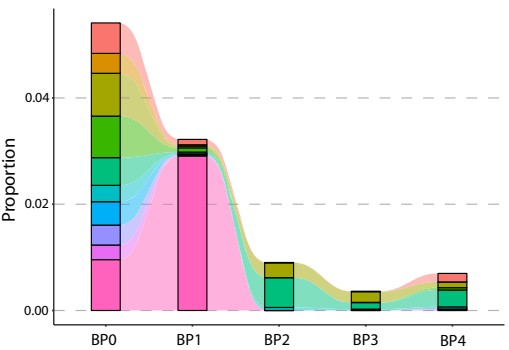

b

### Most abundant clones from BP4 Py117

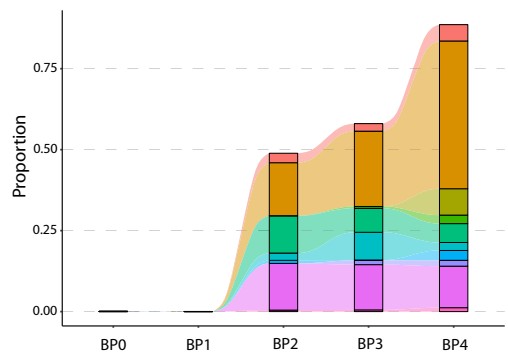

### Most abundant clones from BP4 Py8119

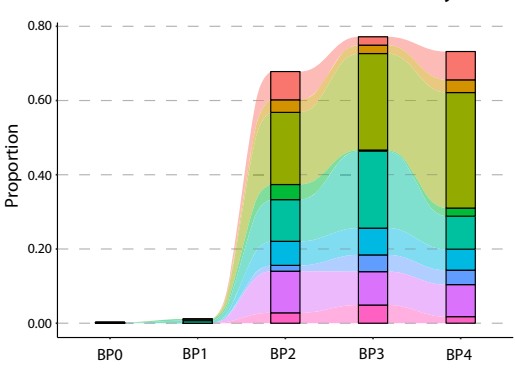

c

### Biopanning: Sanger vs NGS assessment

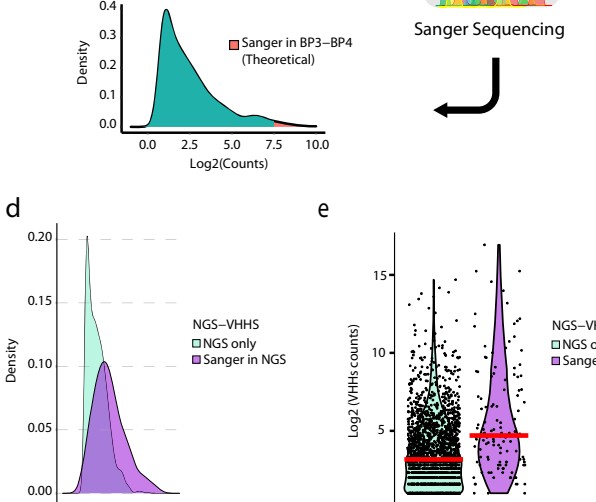

f

### VHH specificity for immune cell subtypes

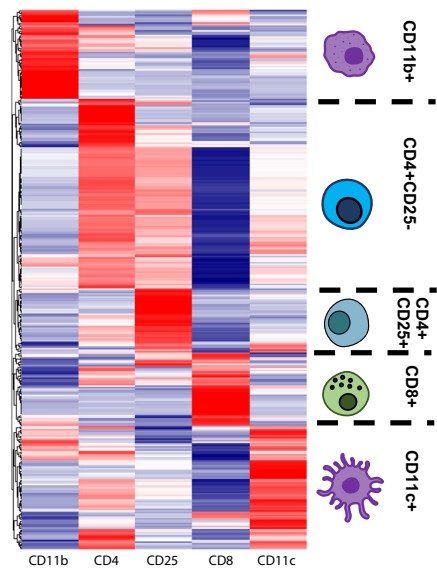

**Fig. 3 | Selection and evaluation of nanobodies developed by in vivo biopanning.** Tracking Top VHHs either in BP0 towards BP4 (**a**) or in BP4 backward to BP0 (**b**) demonstrates the biological enrichment of the biopanning process in vivo. **c** Distribution of VHHs' density after biopanning, where titration followed by Sanger sequencing theoretically provides the most abundant clones, such as the top 25%. and NGS allows for deeper assessment of biopanning with a boarder range of assessment at a multiplexed scale. **d** Experimental distribution of abundance by density of all enriched clones in BP3 and BP4 identified by NGS only or clones also identified by Sanger showing deeper identification of enriched clones where Sanger identified the most abundant clones. **e** Log2 counts of clones identified by NGS in BP3-BP4 from all cell sub-types ($n = 2087$ clones) in compared to clones selected for Sanger sequencing clones ($n = 156$ clones), median represented by red line. **f** Relative enrichment of VHHs in each row across the five cell subtypes from BP3-4 that identify VHHs with cell type specificity.

the y-axis shows that the Sanger clones were enriched but covered the broad range of enriched NGS clones in BP3-4 (Fig. 3d). NGS detected many enriched clones within the abundance range of Sanger sequencing, which supports a deeper assessment of enriched clones. This is critical for parallel enrichment given that important antigen-binding VHHs may show a broad range of abundance when selected in a complex system via in vivo biopanning (Fig. 3e).

Before biologically evaluating VHH nanobodies, we developed a set of criteria to shorten the list of NGS-VHHs with meaningful properties that we can infer from NGS. We collapsed the files from each tumor type, cell subpopulation, tumor, and dLN tissue. We then identified clones that were enriched in BP3 and BP4 over BP1 and BP2. To determine if there was cell type selectivity, we analyzed the filtered clones for higher enrichment in later BP rounds rather than in early rounds for each cell subtype. We generated a heatmap that displays the relative enrichment of all clones in each cell subtype compared to each of the other subtypes, which revealed a list of clones that are specific for each cell subtype and also confirmed that parallel enrichment was achieved (Fig. 3f).

We initially hypothesized that the value of parallel enrichment across tissues and tumor types would enable the identification of a library of nanobodies that distinguish unique features of similar cell types in different environments. This could help distinguish important context specific activity. It could also identify antigens specific for such activities. NGS affords this multifaceted evaluation, so we compared unique and overlapping clones identified in biopanning 3 and 4 in Py117 and Py8119 tumors and dLN's. We identified multiple shared and unique nanobodies for CD45 cells, CD8 and CD11c cells (Supplementary Fig. 6a–c). These data suggest biology-specific evaluation and deep dive could result in important cellular observations. This is a focus of ongoing work and future reports.

## Enriched nanobodies alter signaling of targeted immune cells

We eventually aim to use INSPIRE-seq to develop therapeutically active nanobodies so we next evaluated if nanobodies can functionally alter immune cells in the TME. We performed scRNAseq after injecting library pools to identify transcriptional activity that can be attributed to enriched phage. A summary of the cell sorting and scRNAseq experiment is shown in Fig. 4a. We used treatment-unresponsive Py8119 tumors to determine if enriched phage could alter TIL transcriptomic signaling and potential phenotype. This would be a surrogate for whether VHH expressing phage could impact immune cell function. To develop selective pools of enriched phage, the library from BP1 was injected into mice bearing Py8119 tumors. After 2 h, mice were euthanized, and tumors were collected then dissociated to allow infiltrating CD45$^+$ cells to be isolated via FACS. The CD45$^+$ phages were expanded then injected into a second round of Py8119 tumor-bearing mice. Digested cells were sorted for CD8$^+$ T cells and CD11c$^+$MHCII$^+$CD11b$^-$ (CD11c) DCs. Enriched phage pools from CD45$^+$, CD8$^+$, CD11c, PBS, and Phage without a nanobody were injected into tumor-bearing mice to process cells for scRNAseq.

After scRNAseq was performed, samples were pooled, QC was performed, and cells were annotated for relevant cell subtypes that were successfully extracted from the TME. Twelve clusters were obtained and there was equal representation of each cell type across all samples verifying consistent execution and no major impact on the cell types 2 h after library injection (Fig. 4b, Supplementary Fig. 7a–c, and "Methods"). To identify the global changes in the TME, we calculated the differential gene expression (DGE) between injected samples. We used upregulated genes in CD11c-VHH and CD8-VHH injected tumors to perform Go term analysis. There was enrichment of immune response related terms in both samples such as interferon gamma (IFNγ) response, response to interleukin-1, and cytokine and chemokine mediated pathways in the CD11C-VHH sample (Fig. 4c). In CD8-VHH sample, cell killing, adaptive immune response, and lymphocyte mediated immunity were upregulated (Fig. 4d).

To better understand changes on the cellular level and to confirm the specificity of injected samples to target the cells for which they were enriched, we examined the distribution of DGE in cell subpopulations. The CD8 enriched library led to increased expression of several genes in most of the lymphocyte lineages (CD8, NK, and NKT) compared to the other treatment groups. Alternatively, the CD11c enriched library resulted in strong induction of macrophage, dendritic cell, and neutrophil populations. This shows that enriched nanobody expressing phage can specifically alter targeted cell transcriptional activity in the TME related to the cells for which they were enriched (Fig. 4e).

We then sought to identify what signaling pathways were enriched in the targeted cells, DCs in the case of CD11c (Fig. 5) and CD8 T cells in the case of the CD8 enriched library (Fig. 6). To interrogate samples injected with the CD11c enriched library we identified three DC subpopulations (Fig. 5a). When evaluating these specific populations, we noted that in addition to transcriptional and translational pathways, there was class II antigen processing and presentation, IFNγ response, and IL-4 response enrichment (Fig. 5b). In addition, there was significant increase in DC activation signaling in CD11c and CD8 samples as compared to the CD45 sample ($p = 5.21e−03$ and $p = 0.01$, respectively) or as compared to PBS sample ($p = 0.01$ and $p = 0.02$, respectively) (Fig. 5c). We next interrogated genes involved in DC function including Type 1 interferon signaling, DC maturation, and DC regulation and observed increase expression in cells from the CD11c and CD8 phage samples (Fig. 5d). We confirmed the source of these genes were from the DC sub-populations (Fig. 5e).

Alternatively, the CD8 T cells showed IFNγ response, antigen processing, cell killing, lymphocyte mediated immunity, cellular response to IFNγ, adaptive immune response, and T cell mediated immunity in the enriched CD8-VHH injected sample (Fig. 6a, b). We sought to evaluate changes in CD8 T cell sub-populations and observed five distinct sub-populations (Fig. 6c). Cluster 5 and to a lesser extent 0 and 3 were overrepresented by T cells from the CD8-VHH (70%, 49.7%, and 34.8%, respectively) and CD11c-VHH (25%, 27.7, and 25.8, respectively) enriched libraries (Fig. 6d). We then identified that these clusters, especially cluster 5, had prominent expression of immune checkpoints, cytokines and effector molecules, co-stimulatory molecules, transcription factors, and little to no expression of naïve markers (Fig. 6e). This signaling in these clusters was greater in in the CD8 and CD11c enriched samples (Fig. 6e). To confirm these changes, we used the DGE between CD8 T cell sub-populations to examine the Go term enrichment in subcluster 5. We found that subcluster 5 was enriched in adaptive immune response, T cell differentiation and activation, T cell proliferation, and cell killing, which coveys there was activation of the targeted CD8 T cells by the CD8-VHH enriched library (Fig. 6f).

## Dendritic cell nanobodies bind target cells variably across tissues

Our next goal was to evaluate if enriched nanobodies from in vivo selection bind primary cells, and to then identify target antigens. Considering the long-term goal of developing nanobodies that can modulate immune responses, we focused our attention on antigen-presenting DCs, which are necessary for developing adaptive immunity. Many innate immune receptors on DCs can mediate maturation, adaptive immunity, and regulatory activity, making them a critical target[22]. They can be rendered inactive within the harsh TME, thereby requiring innate agonists and cytokine signaling to develop antibody-based immune responses[23]. More recently, monocytic DCs have been shown to sustain T cell function but are susceptible to cell-initiated death via lysosomes. Conventional DCs may act with regulatory functions limiting signaling of adaptive immune responses[24,25]. Therefore,

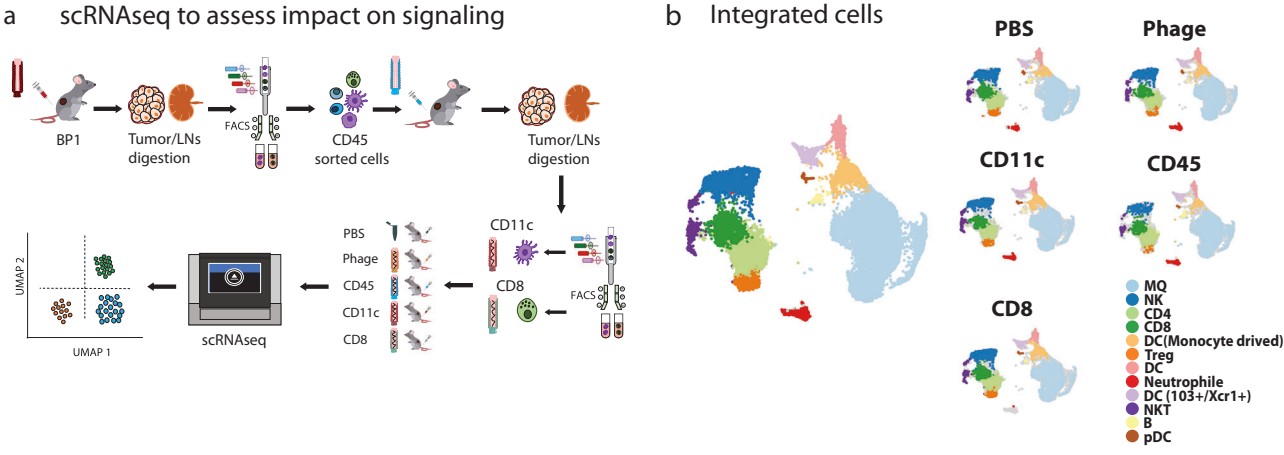

a    scRNAseq to assess impact on signaling

b    Integrated cells

c    GoTerm of all cells in CD11c-VHH injected library

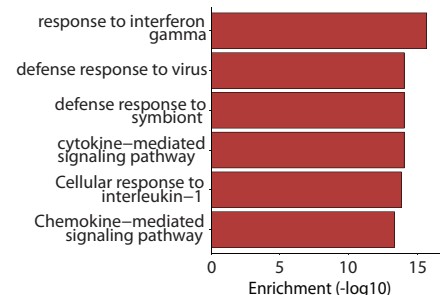

d    GoTerm of all cells in CD8-VHH injected library

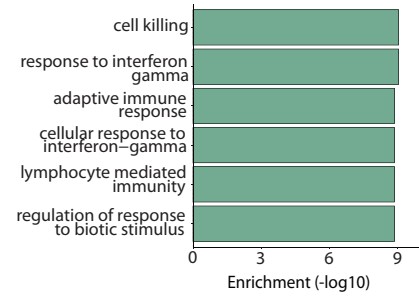

e    Specificity of injected VHHs libraries to change signaling of targeted sub-populations

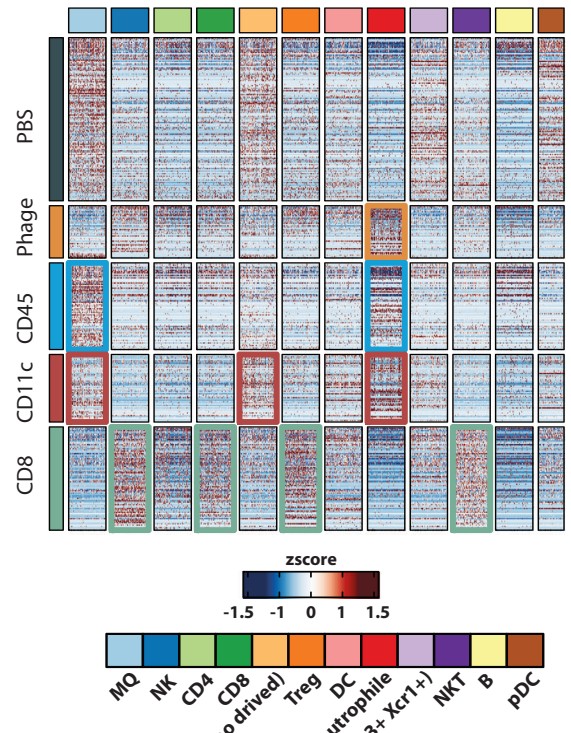

**Fig. 4 | ScRNAseq to explore functional changes introduced by VHHs in target immune cells. a** Schema for scRNAseq experimental outline to determine functional changes in target immune cell populations. **b** Cell populations across all samples showing harmonized and integrated samples (left), cells clustered by sample type showing comparable distribution of cell populations regardless of the samples (right). **c, d** GO term enrichment of upregulated genes in CD11c-VHHs and CD8-VHHs library injected mice respectively were performed using enricher package, significance was determined by Fisher exact test (−log10) of adjusted *p* value by benjamini−hochberg (BH) method. **e** Heatmap showing differential gene expression analysis (DGE) between samples (rows) and their expression distribution across cell populations (columns). Colored boxes highlighted cell subpopulation that showed most DEGs in the respected sample (rows). For example, in CD11c sample most altered cell populations were DC (Monocytes derived) and Neutrophil (red boxes). In the CD8 sample, the most altered cell populations were NK, CD8, Treg, and NKT (green boxes). Source data are provided as a Source Data file. Mouse illustration was adapted https://creazilla.com/nodes/7772730-mouse-clipart under creative commons license (CC0).

we selected 15 enriched clones from high dilution, then expressed and purified the proteins. Each clone was incubated with splenocytes, then counter-stained with an anti-His secondary antibody to determine selective binding. We cloned two controls, a DC-binding VHH (DC2.1) and a negative control VHH (BCII10)[26]. We identified four nanobodies that had strong binding to CD11c⁺CD11b⁻ splenocytes then cloned the nanobodies into Venus fluorescent fusion protein vectors (Fig. 7a). After nanobody purification, gel electrophoresis verified purity and

fusion protein-maintained fluorescence (Ven-Nb1-4). A separate western blot further showed intact nanobodies using an anti-His antibody (Fig. 7b).

We next evaluated if fusion proteins-maintained binding to immune cells extracted from the spleen, lymph node, and tumor. Py117 tumor cells were injected into naïve mice, and cells were extracted for flow cytometry analysis from each tissue to identify the cell types used in selection; CD11c⁺CD11b⁻, CD11c⁻CD11b⁺, CD8⁺, and CD4⁺ cells, can be

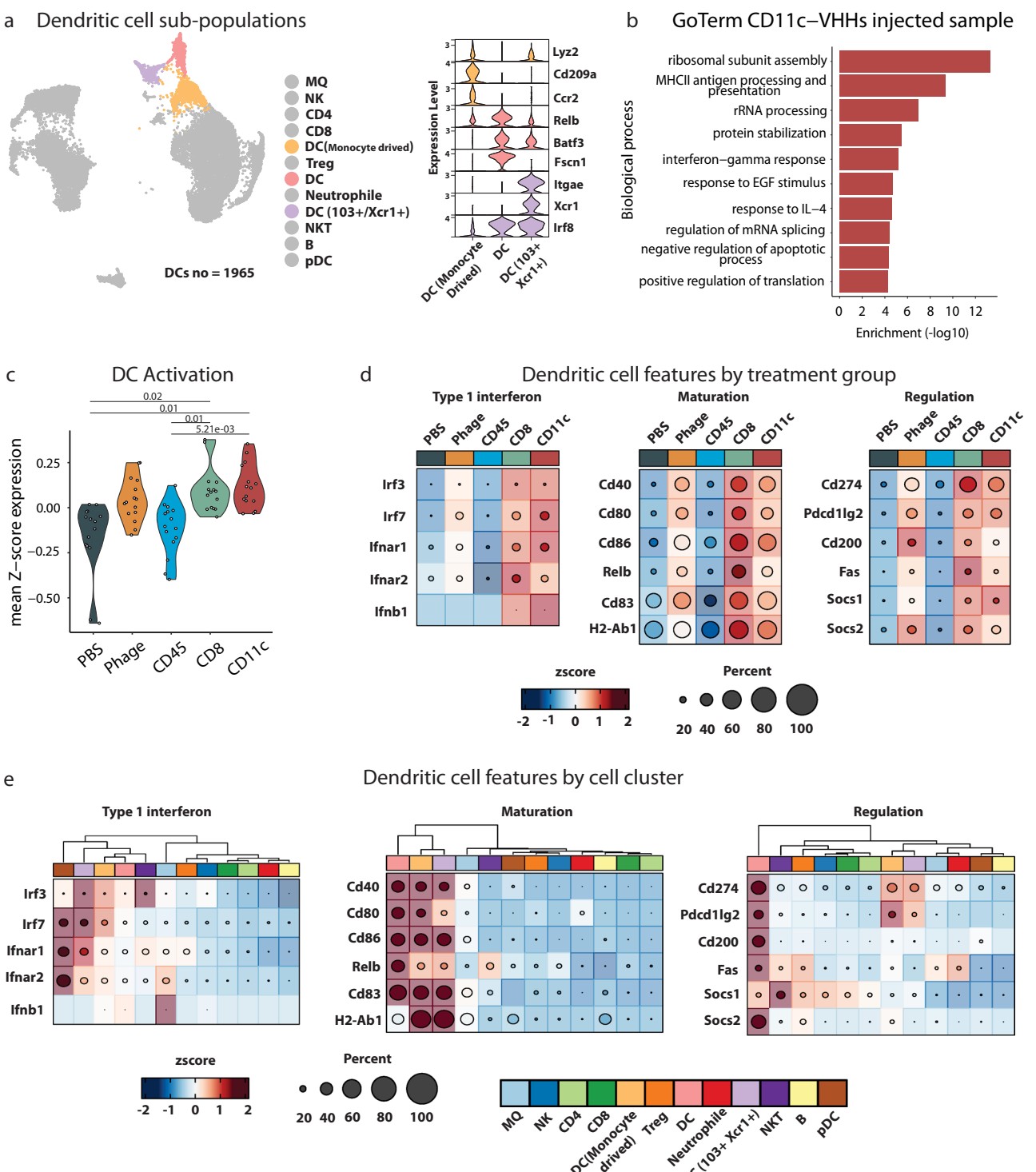

**Fig. 5 | Transcriptional changes introduced by CD11c VHHs injected library.**
**a** DC populations and their example of canonical markers used in the annotation process. **b** GO term enrichment of upregulated genes in CD11c-VHHs library injected mice in DCs subpopulations were performed using enricher package, significance was determined by Fisher exact test (−log10) of adjusted *p* value by benjamini-hochberg (BH) method. **c** Mean expression of DC activation GO term genes identified in VHHs library injected mice (F Welch test with Holm adjusted *p* value). **d**, **e** Heatmap showing Type 1 interferon, maturation, and regulations genes expression in all samples and cell subpopulations respectively. Source data are provided as a Source Data file.

seen in Fig. 7c. The DC2.1 and BCII10 were shown to be effective controls and we identified differential binding of the Ven-NB1-4 on dendritic cells (CD11c⁺CD11b⁻) in the TME (Fig. 7d). Although the target antigens were unknown, we hypothesized that the nanobodies could have differential binding to DCs in the tumor, lymph node, and spleen, given the fact that differential cell states exist in each organ (Fig. 7d).

We observed that Ven-NB1 bound efficiently to tumor and lymph node DCs but not splenic DCs. Alternatively, as a fusion protein, Ven-Nb2 only bound DCs in the tumor. Ven-Nb3 had high binding in all tissues and bound to a subset of splenic cells. We chose to pursue Nb1 for target identification due to tumor and node binding, which may differentiate functional DCs.

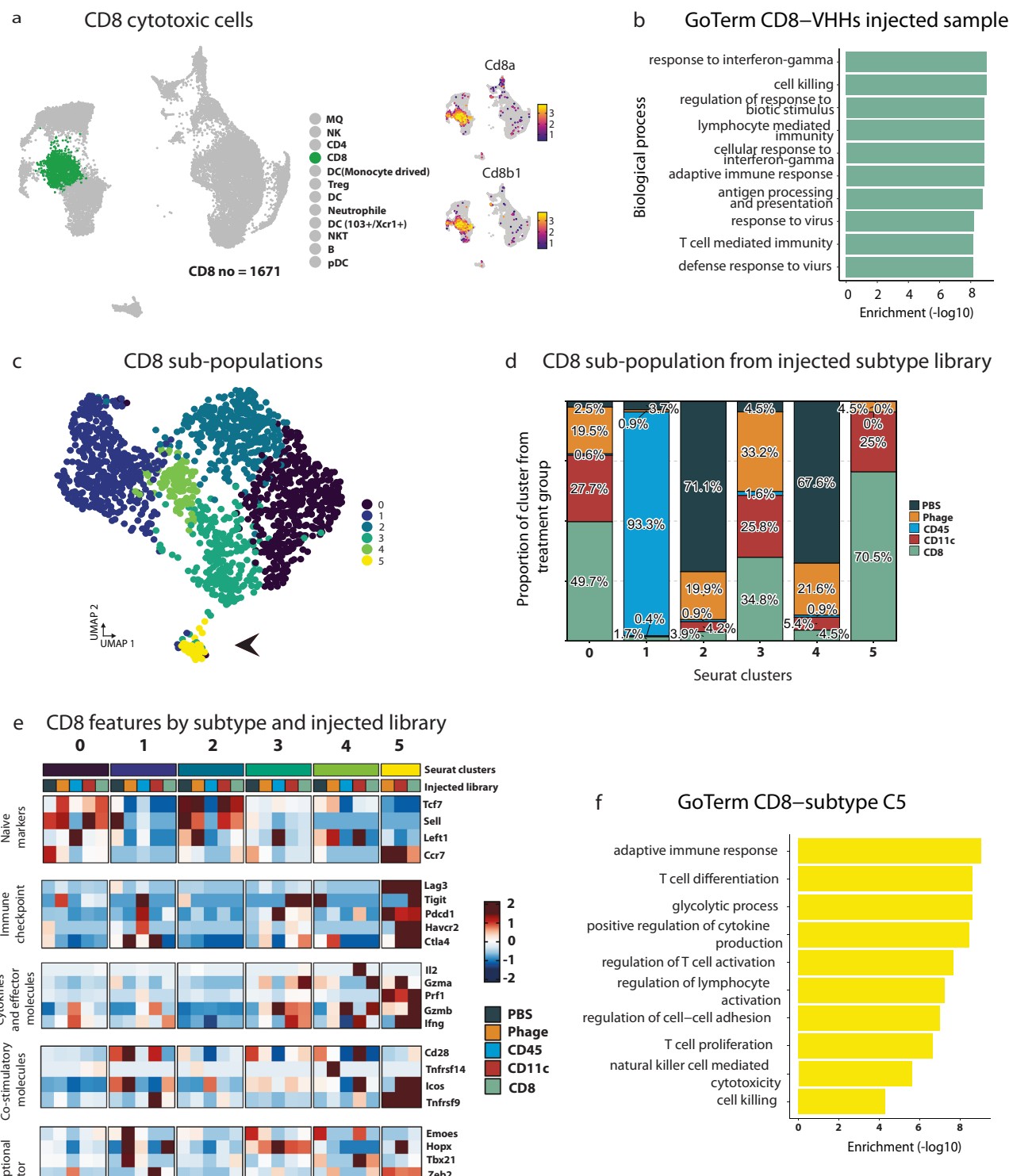

**Fig. 6 | Transcriptional changes introduced by CD8 VHHs injected library.**
**a** CD8 T cells and expression of their canonical markers. **b** GO term enrichment of upregulated genes in CD8-VHHs library injected mice in CD8 T cells were performed using enricher package, significance was determined by Fisher exact test (−log10) of adjusted p value by benjamini-hochberg (BH) method. **c** UMAP plot showing the CD8 sub-populations by sub-clustering CD8 T cells. Black arrows refer to sub-population cluster 5. **d** Histogram showing relative distribution of CD8+ T cell sub-populations based upon sample type. **e** Heatmap showing naïve markers,

immune checkpoint, cytokines and effector molecules, co-stimulatory molecules, and transcriptional factor genes expression in CD8 T cells sub-populations and injected libraries. **f** GO term enrichment of upregulated genes in CD8-VHHs library injected mice in CD8 T cells cluster 5, enrichment was performed using enricher package, significance was determined by Fisher exact test (−log10) of adjusted *p* value by benjamini-hochberg (BH) method. Source data are provided as a Source Data file.

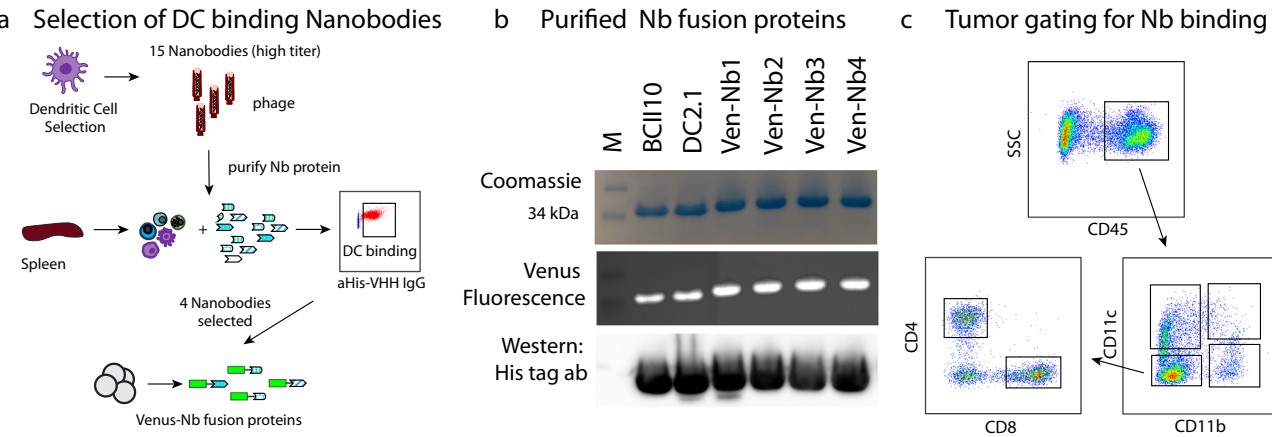

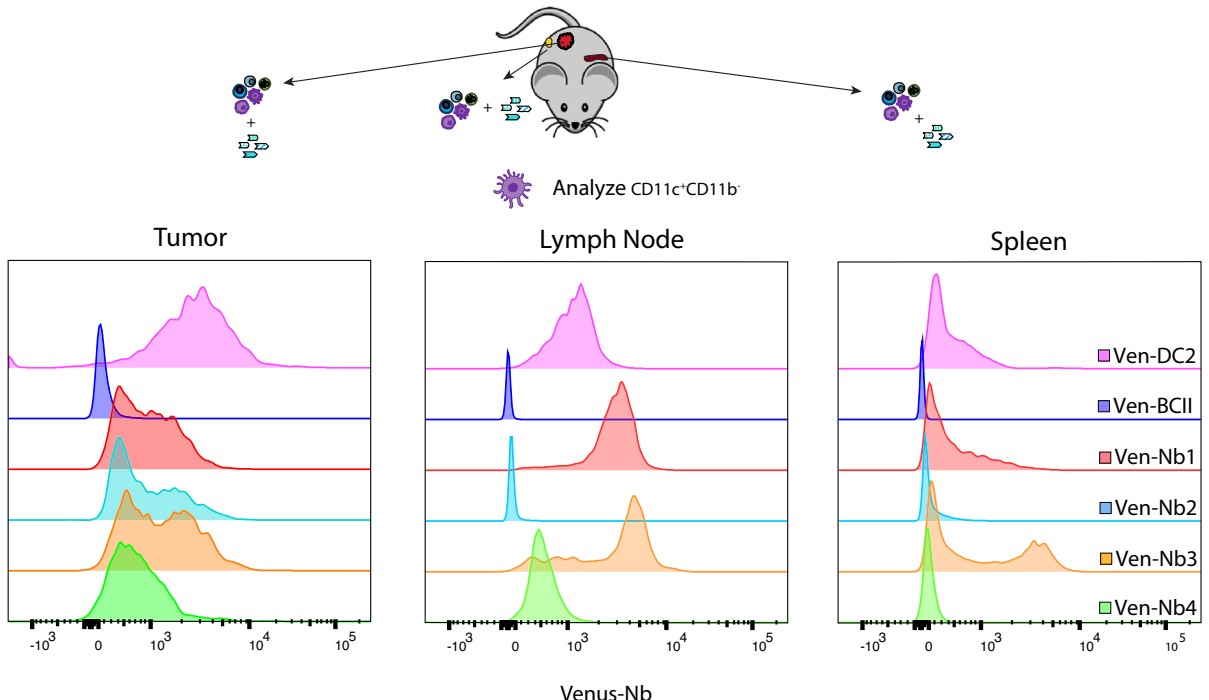

**Fig. 7 | DC-VHHs selection and differential binding of VHHs across immune cells and tissues. a** Pipeline to select VHHs that preferably bind to dendritic cells and then generate Venus-VHH fusions. **b** Purified fusion protein by ÄKTA pure detected by Coomassie blue staining, Venus fluorescence, and Immunoblot using α−His antibody (representative of 3 independent experiments). **c** Flowcytometry gating strategy to select cells sub-populations CD11c⁺, CD11b⁺, CD8⁺, and CD4⁺.

**d** Detection of the selected Venus fusion DC-VHHs, BCII10-VHH, and DC2.1-VHH in the tumor, spleen, and LN of Py117 tumor-paired mice (representative experiment) from three independents experiments. Source data are provided as a Source Data file. Mouse illustration was adapted https://creazilla.com/nodes/19275-cartoon-grey-mouse-clipart under creative commons license (CC0).

## The target antigen for dendritic cell selected Nb1 is PHB2

After showing the robustness of the INSPIRE-seq pipeline and that phage pools can alter cell signaling, we next sought to identify one of the antigens for a nanobody with DC selectivity. Immunoprecipitation and mass spectrometry experiments were performed by first generating FC-Nanobody fusion protein constructs to improve valency for pull-down and to avoid non-specific binding with secondary antibodies. We then expressed and purified the IgGFC-Nb1 protein. Protein G-coated dynabeads were blocked with splenocyte membrane protein preparations, loaded with the IgGFC-Nb1 protein, with immunoprecipitation finally performed using pre-cleared splenocyte membrane extracts (Fig. 8a). Three independent pull-downs using the IgGFC

fusions revealed a band that was observed only in the presence of IgGFC-Nb1 with splenocyte membrane proteins (Supplementary Fig. 8a). The band of approximately 33 kDa band was extracted and analyzed by tandem mass spectrometry. PHB2 protein was the most abundant protein in each of the replicates (Supplementary Fig. 8b). We subsequently performed a western blot for PHB2 by using the products after pull-down. A strong band consistent with the protein that was only in the preparation with IgGFC-Nb1 and splenocyte membrane proteins was observed, which was not observed with IgGFC-Nb2 (Fig. 8b).

PHB2 has multiple roles based on cell-compartment and tissue specificity, including cell membrane and cell signaling functions. *PHB2*

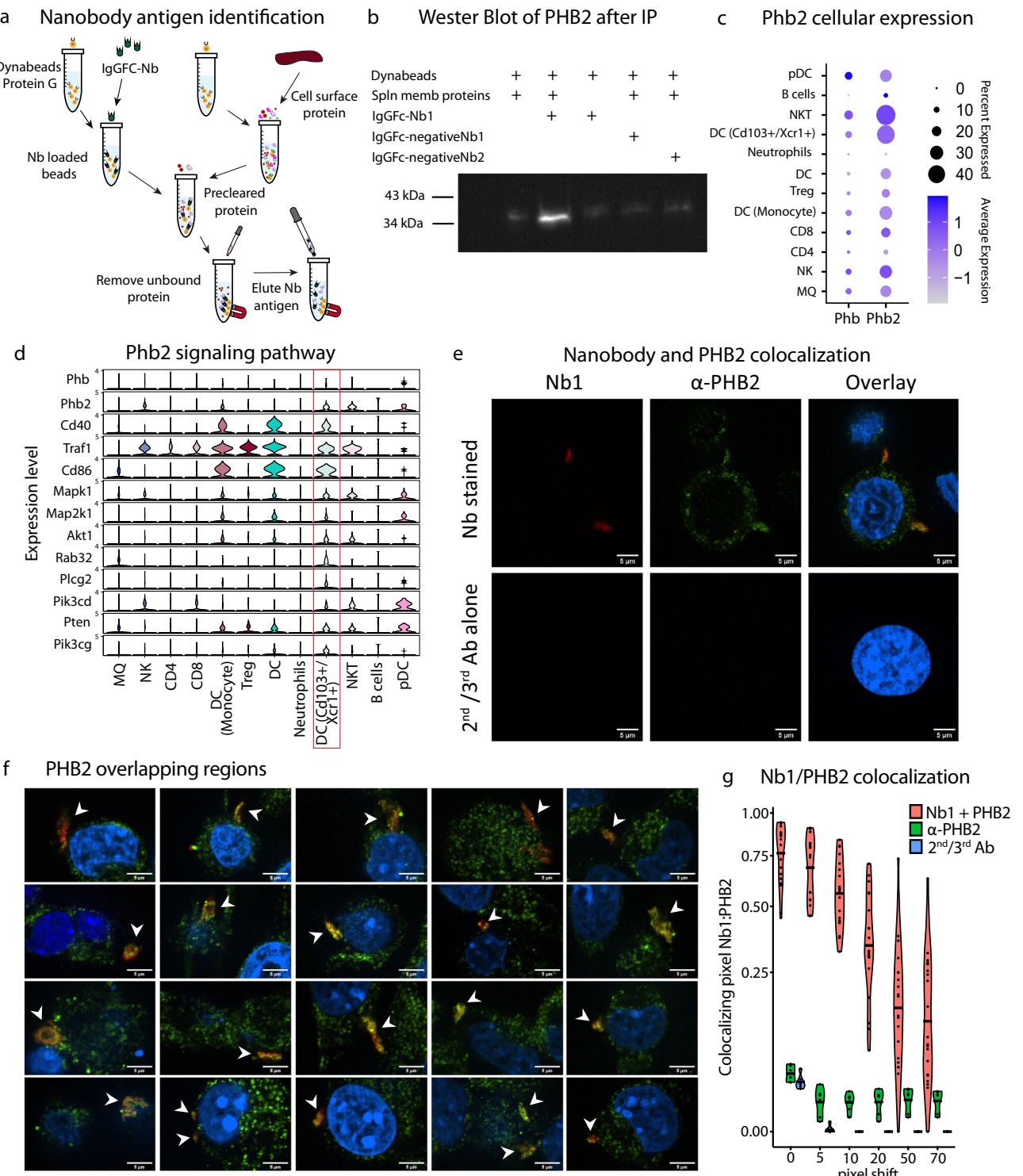

**Fig. 8 | Identification and verification of Nb1 binding to Phb2. a** Nanobody pull-down and antigen identification pipeline using immunoprecipitation. **b** Western blot of α-PHB2 protein after pulled-down by Nb-FC fusion protein (IgGFC-Nb1) (representative of 3 independent experiments) **c** Dot plot for Phb and Phb2 expression across immune cell populations. **d** Violin plot showing the Phb2 signaling pathway across immune cells populations, revealing the greatest expression in the DC CD103[+]/Xcr1[+] population (red box). **e** Representative confocal images from two independent experiments of Nb1 and PHB2 colocalization in MC38 cells with the upper row Nb1/α-PHB2 stained and lower row secondary and tertiary antibody alone control. The first column is the nanobody channel red, then α-PHB2 green, and the overlay with Höchst 33342 blue (scale 5 μm).
**f** Representative confocal images from two independent experiments of Nb1 colocalization with membranous regions expressing PHB2 from 20 different cells (white arrows). **g** Mander's colocalization coefficient M1 of Nb1 pixel overlapping PHB2 pixel (mean represented by black line) for Nb1 and α-PHB2 ($n = 20$ images), α-PHB2 antibody alone ($n = 6$) and 2nd/3rd antibodies alone ($n = 6$) from two independent experiments with six incrementing pixel shift configurations. Source data are provided as a Source Data file.

cooperates with CD86 on the cell membrane of B lymphocytes to regulate IgGI production levels and upon CD40 engagement, is required to activate NF-κB signaling[27]. It is also one of the genes in the GO term for DC activation, thus, the role on DCs could be relevant to immune activation during immune responses. We sought to determine if there were differences in immune activation genes associated with *PHB2* gene signaling from the scRNAseq experiments described in Fig. 4. We evaluated both *PHB1* and *PHB2* expression across cell types of all scRNAseq samples and noted that *CD103⁺/Xcr1⁺* DCs were among the cells with the greatest expression (Fig. 8c). We then evaluated genes in the *PHB2* signaling pathway and observed that the *CD103⁺/Xcr1⁺* DCs most consistently had the highest expression of these genes, suggesting that *PHB2*-related signaling in the DCs was intact after DC library injection (Fig. 8d). We went back to Fig. 3f and noted that the PHB2 binding phage was one of the clones in the CD11c⁺ cells that also had strong binding to CD4⁺ T cells.

Since Nb1 appears to bind to PHB2, we sought to confirm this by colocalization experiments with immunofluorescence. In these experiments we fixed tumor cells that had strong expression and stained them with α-PHB2 antibody as well as Nb1. Then, we used a secondary antibody for α-PHB2 and α-VHH antibody, along with a tertiary antibody for the α-VHH. Cells were imaged by confocal light microscopy and we observed that not all the PHB2 signal colocalized with Nb1, but that nearly all of the Nb1 signal colocalized with PHB2. The colocalized staining occurred along the cellular membrane in dense regions (Fig. 8e). This high degree of colocalization was observed in many cells with the membranous subunits across repeat experiments (Fig. 8f). Colocalization was quantified by Mander's colocalization coefficient M1 as the proportion of Nb1 pixels that overlap PHB pixels. We incorporated a pixel shift to show colocalization significance, since with image shifting there is a decreased proportion of colocalized pixels (Fig. 8g)

There is much interest in whether Nbs discovered through INSPIRE-seq could translate to human protein targets, thus having cross-species reactivity. The PHB2 human protein sequence is identical to the murine protein so modeling discussed in the next section would be identical. Therefore, we stained human cells with PHB2 antibody and Nb1 to determine if there was cross reactivity. We observed the same pattern of staining on the cellular membranes and as the mouse cells (Supplementary Fig. 9a). Mander's colocalization coefficient verified colocalization (Supplementary Fig. 9b).

### Binding interface modeling aids nanobody selection and validation

The INSPIRE-seq pipeline should be accelerated with computational means to assess binding interactions and to facilitate making binding predictions one day. With this in mind and to gain more confidence in PHB2 binding, we sought to nominate the binding interface between Nb1 and PHB2. We performed computational docking using the Rosetta Suite for protein modeling and design using AlphaFold structure predictions[28–30]. The protocol overview is shown in Fig. 9a. We used the Rosetta Antibody application to model the atomic coordinates of Nb1 from its amino acid sequence[31,32], first using homology modeling of the highly conserved nanobody framework, CDR1 and CDR2, and then using loop modeling for the variable CDR3[33]. A total of 10,000 Nb1 models were generated in 37 clusters according to their conformation and total energy, a value indicative of overall stability (Fig. 9b)[34]. The representative Nb1 models for the eight lowest-score clusters (red circles in Fig. 9b) were selected as top candidates for molecular docking. Their tridimensional structure is shown in Fig. 9c.

The PHB2 structure has been only partially determined experimentally by X-Ray crystallography (PDB-ID 6IQE). It shows that residues 187 to 245 are involved in an anti-parallel coiled coil motif[35]. The experimental structure was used to generate PHB2 5x_CC, a 5x coiled coil model to mimic the minimal unit for the multimeric state of the

protein (Supplementary Fig. 10a). We then obtained the full model of PHB2 from the AlphaFold database (Supplementary Fig. 10b)[36], which is composed of a minimal intracellular domain, a single helix transmembrane region, and an extended extracellular portion composed of an N-terminal, coiled coil and C-terminal domains. In AlphaFold, the N-terminal and coiled coil domains are predicted with high accuracy, but the C-terminal domain is not.

The docking protocol consisted of two steps, as shown in Fig. 9a: global docking, to randomly test different orientations in the interaction between Nb1 and PHB2[32,37,38], and SnugDock, a local docking protocol optimized to include backbone flexibility in the CDRs of antibodies and nanobodies[39,40]. Among the 28,874 models generated with global docking, 71 clusters were identified (cyan and blue circles in Fig. 9d). Cut-off values of −30 Rosetta Energy Units (REU) in binding energy and −1175 REU in total energy were used to select 11 Nb1-Target pairs for the next step of local docking with the SnugDock protocol (blue circles in Fig. 9d). Docking funnels for all eleven initial docking sites are shown in Supplementary Fig. 10c. We considered sites 1, 5, 6, and 8 for further analysis since the binding strength of −40 REU or lower was achieved with at least ten models each (Fig. 9e). After analysis with the 5x_CC variant, which represent the hypothetical multimeric state of PHB2, docking sites 5 and 6 were discarded due to clashes between Nb1 and alternative chains of PHB2 (Supplementary Fig. 10d). Docking site 1, targeting the coiled coil and N-terminal domains (Fig. 9f, left), and docking site 8, targeting the coiled coil and the C-terminal domains (Fig. 9f, right), represent the most realistic binding sites according to computational modeling and docking, and will be considered for further analysis.

## Discussion

In this study, we demonstrated that INSPIRE-seq, which utilizes phage display technology, can identify cell binding VHH nanobodies through in vivo selection designed to identify targets and nanobodies that bind to TILs. By using NGS to evaluate biopanning of selected immune cell sub-populations, there is robust enrichment of nanobodies for immune cells in the TME. Not only did our screening identify enrichment of the broad CD45⁺ population, but it also identified parallel enrichment in which multiple nanobodies selective for different cell subtypes could be identified. The power of this approach was amplified using NGS assessment, which brought us to the conclusion that fewer rounds of biopanning are necessary to achieve enriched phage, thereby allowing for screens in complex experimental settings. We believe that these methods could be broadly applicable to multiple biologic discovery experiments in different pathologic contexts. We hypothesize that these methods can be extended to study complex in vivo microenvironments and identify nanobodies that are useful for therapeutic and diagnostic applications, as suggested by the scRNAseq experiment. We next aim to demonstrate that this technological platform can be used to develop novel cancer immunotherapies.

Many traditional methods study the biology of cells or tissues to identify and validate targets while other approaches perform screening in a context or environment not native to an IV-delivered drug. INSPIRE-seq seeks to (1) identify the targeting moiety of a future drug or target to be studied biologically and (2) select it by the ability of the viral particle to home to the target in vivo. Alternatively, scRNAseq technology provides a potent way to study TME and nominates therapeutic targets and biomarkers in multiple settings. However, the typical scRNAseq pipeline required cell dissociation that could alter cell transcriptomic composition[41,42]. Moreover, single-cell proteomics technology is still under development, but will revolutionize our understating of cell-cell communications once fully achieved[43]. In this work and in the coming years, we will demonstrate that INSPIRE-seq offers a significant step forward to achieve single-cell proteomic selection through unbiased scanning to identify antigens in an intact environment. We observed that INSPIRE-seq can in real-time capture

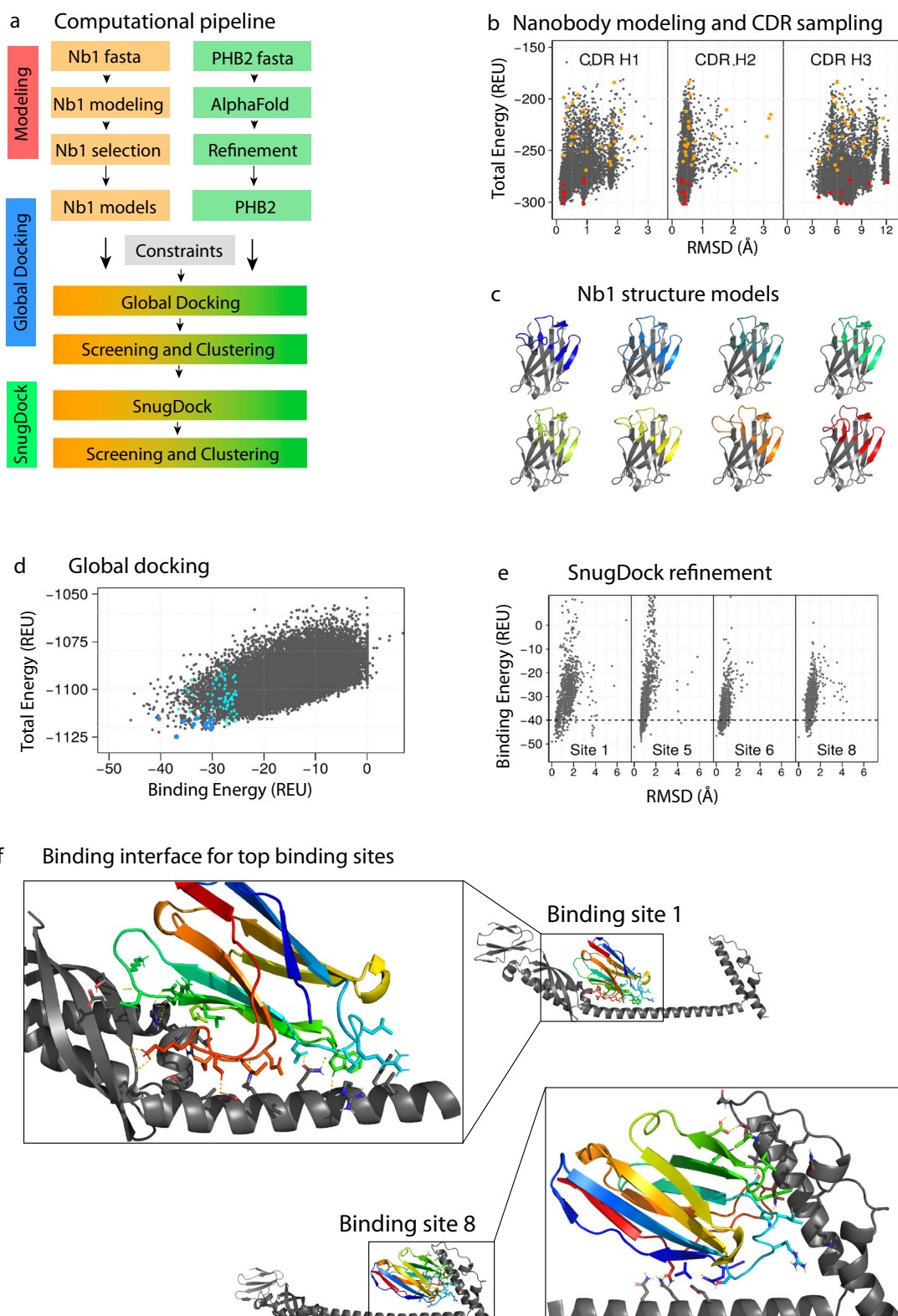

**Fig. 9 | Computational modeling for docking predictions of Nb1 to PHB2.**
**a** Computational protocol for modeling and docking using Rosetta and AlphaFold.
**b** Modeling of Nb1: Rosetta total energy vs. RMSD for each individual CDR. The best representative model for each cluster is shown in orange circles, and the top eight are shown in red. **c** Cartoon representation of the eight selected Nb1 candidates for computational docking. **d** Rosetta total energy vs. Rosetta binding energy after global docking.

global docking. The best representative model for each cluster is shown in cyan circles, and the top 11 are shown in blue circles. **e** SnugDock binding energy vs RMSD results for binding sites 1, 5, 6 and 8 (full results in Supplementary Fig. 7e). **f** Cartoon representation of binding sites 1 (left) and 8 (right). PHB2 is shown in gray, while Nb1 is shown with rainbow color. The residues forming critical interactions are shown as sticks, and hydrogen bonds are shown as yellow dash lines.

changes in cell-cell communications, therapy response dynamics, and was less sensitive to perturbations introduced by the methodology, which we believe is advantageous compared to scRNAseq and other approaches.

Our scRNAseq experiment showed that INSPIRE-seq-enriched VHHs altered the functionality of cells and underling singling pathways. Not only can INSPRE-seq be used for drug development through optimizing the targeting moiety or modifying the nanobody to make it a drug, but nanobodies can activate and block signaling of targets. We can now evaluate and select VHHs that actively alter the TME and cell-cell communications. Our future aims will further develop approaches to unlock the opportunities for selecting functionally active nanobodies.

A major challenge, but opportunity for INSPIRE-seq, is that it is an in vivo selection strategy using a diverse library of unknown nanobodies, performs selection in a complex microenvironment, and identifies nanobodies that bind to unknown targets on cells. We believe and aim to show that with new technologies, this selection strategy has strong potential despite the fact that targets are unidentified at the time of selection. Technological innovations that can further our aims include machine learning advances in three-dimensional protein modeling. Recently, AlphaFold and RoseTTA-Fold achieved major scientific breakthroughs in terms of making accurate predictions quickly. As shown in this study, these models using Rosetta tools can propose nanobody and target binding interfaces. We intend to couple INSPIRE-seq with these technologies to predict antigens that our NGS-identified VHHs will bind to prior to immunoprecipitation and mass spectroscopy, as we did with Nb1 in Fig. 8. Though it is a difficult task, the pieces are in place to begin this effort through carefully matching wet and dry lab validation. This will allow us to precisely select the VHHs of interest thereby accelerating selection and application.

## Methods

This research complies with all relevant ethical regulations including NIH guidelines. The ethical oversight of laboratory animal use is governed the University of Texas Southwestern Medical Center Institutional Animal Care and use Committee (IACUC) and all protocols are approved.

### Chemicals and biological materials

The nanobody-encoding M13 phage display library was purchased from Abcore Inc. (Ramona, CA). TG1 electrocompetent cells were procured from Lucigen (Middleton, WI). M13KO7 helper phage particle, restriction and modification enzymes, and PCR amplification materials were from New England Biolabs (Ipswich, MA). Antibiotics and bacterial culture media such as 2-YT broth, Terrific broth, and Agar were from ThermoFisher Scientific. All fine chemicals, IPTG, PEG8000, and Imidazole were from Millipore Sigma (St. Louis, MO). HisTrap columns and Ni sepharose columns were from GE Healthcare Life Sciences (Pittsburgh, PA). Cell culture materials were from Mediatech Inc. (Manassas, VA) and Cytiva HyClone Fetal Clone II was purchased from GE Healthcare Life Sciences.

### Nanobody phage library preparation

The llama VHH antibody library was purchased from Abcore Inc. (Ramona, CA). The library was generated using RNA isolated from peripheral blood mononuclear cells (PBMC) cells that were collected from 20 naïve (non-immunized). A combined total of $1.5 \times 10^9$ PBMC cells (approximately $1 \times 10^8$ per llama) were isolated for RNA production. Total RNA was purified with phenol/chloroform extraction, followed by silica-spin column method. Total RNA was eluted with RNase-free $H_2O$. Quality of RNA was evaluated by OD260/280 ratio (>1.9) and agarose gel electrophoresis (non-denaturing). RNA concentration was estimated using formula of 1.0 OD260 = 40 μg/ml. Library

construction was done by QooLabs Inc. (Carlsbad, CA). Reverse transcription and a primer specific for llama IgG was used to prime the total RNA to generate full length cDNA. The quality of cDNA was evaluated by PCR using llama IgG heavy chain specific primers spanning the variable region and the constant region. Products of VH and VHH with expected sizes were amplified from the cDNA using primers to enable cloning into phage display vector pADL20. Products of VHH were further purified and modified with sfiI sites for cloning into the pADL20c from Antibody Design Labs (ABDL, San Diego, CA) phage display vector. The ligated DNA was then transformed into TG1 cells. A total of $2 \times 10^9$ independent clones were obtained for the library. Phage were then amplified to generate phage lysates with a titer of $2.5 \times 10^{11}$. The VHH phage library was quality controlled by rescue using helper phage VCSM13. One hundred independent clones were selected randomly, and DNA inserts of each clone were sequenced. Over 90% of the clones represent putative immunoglobulin sequences in correct length and reading frame.

### Cell culture

Py8119 and Py117 breast cancer cell lines were derived from spontaneous tumors of transgenic MMTV-PyMT mice congenic in the CL57BL/6 background and were provided by Dr. Ellies[20,44]. Cells were cultured and maintained in Hams F12 media (Corning) and supplemented with 5% Cytiva HyClone Fetal Clone II (GE Healthcare Lifesciences), MITO+ (1:1000 dilution, BD Biosciences, CA), 50 mg/ml gentamicin and 2.5 mg/ml amphotericin B. For tumor implantation, cells were seeded in a complete medium 48 h before implantation at 60–70% confluence. Fresh medium was changed 24 h prior to harvesting for implantation. Py8119 and Py117 cells with a passage number ranging from 15 to 25 were used for all tumor implantations. MC38 colorectal cancer cells were provided by Dr. Engleman and cultured in DMEM (Corning) supplemented with 10% Cytiva HyClone Fetal Clone II. H1299 human non-cell lung carcinoma cells were provided by Dr. Story and cultured in RPMI 1640 (Corning) supplemented with 10% Cytiva HyClone Fetal Clone II. Both MC38 and H1299 cells with a passage number between 4 and 20 were used for immunofluorescence microscopy experiments. Cells were tested for mycoplasma contamination periodically using ATCC PCR Mycoplasma Detection Kit (Manassas, VA).

### Animals

All biopanning experiments were carried out using C57BL/6 wild-type mice with age groups ranging from six to eight weeks. All female mice were used for breast cancer tumor models given the sex of the donor cells and a female breast cancer predominance of approximately 99%. All animals were purchased from Jackson Laboratories (Bar Harbor, ME), and maintained in UT Southwestern Medical Center animal facilities by following guidelines according to UT Southwestern Medical Center Institutional Animal Care and Use Committee (2017-102240). All animal procedures were conducted according to the NIH guidelines for the care and use of laboratory animals and biological safety. Animal protocols were approved by the University of Texas Southwestern Medical Center Institutional Animal Care and use Committee (IACUC) under protocol number 102240 and the facility is AAALAC accredited. Mice were housed with a standard day dark/light cycle 6:00 a.m. to 5:59 p.m., housed at ambient temperature, and humidified ventilated air. Tumors were implanted subcutaneously in the mammary fat pad with $1 \times 10^6$ cells of Py8119 or Py117 breast cancer cells suspended in 50 μl PBS and mixed with an equal volume of growth-factor reduced Matrigel (Corning-356231). Mice would be euthanized with a maximal tumor size of 17.5 mm in largest diameter, but experiments were performed when tumors were 5–12 mm diameter. Experiments with Py8119 or Py117 tumor-bearing mice were conducted two or three weeks after inoculation, respectively. The phage library was injected intravenously in 150 μl of PBS pH 7.2 and

animals were monitored for 2 h following the post-injection period. We noted that injection of undiluted and high titer ($10^{14}$) phages reduced mice activity. However, with a 100-fold dilution, there was no clear physiologic effect on the animals.

## Isolation of immune cells

Mice were euthanized 2 h post IV injection. LNs and tumors were excised, and tissue was dissociated into a single-cell suspension. The LN was processed by gentle homogenization and 40 µm filtration in 10% serum-supplemented medium. Tumor tissue samples were digested with 4 ml of serum-free medium containing 5 µM liberase and 100 µM DNAse for 40 min at 37 °C with shaking at 70 RPM. Tumor digestion protocol #4 was performed by using the gentleMACS Octo tissue dissociator (Miltenyi Biotec) with three 10 min rounds of processing over 40 min. Serum-supplemented medium was added to the digestion and cells were filtered through a 40 µm filter. The cell suspension was pelleted, then RBCs were lysed for 5 min at 4 °C in ACK buffer. Cells were then resuspended in fresh medium. TILs were isolated via ficoll gradient, suspended in fresh medium, and washed. Cells were resuspended in MACS staining buffer and passed through LS columns with a dead cell removal magnetic bead kit (Miltenyi Biotec). CD11b⁺-positive selection magnetic beads (Miltenyi Biotec) were first used to isolate CD11b⁺ cells. The flow through was then stained with CD11c+ positive selection magnetic beads (Miltenyi Biotec) to isolate CD11c⁺CD11b⁻ cells. The flow through cells were then divided, 1/3 for CD8⁺ selection (Miltenyi Biotec) and 2/3 for CD4⁺ subsets (both positive selection for CD4⁺CD25⁺cells and negative selection for CD4⁺CD25⁻ cells Miltenyi Biotec). A small portion of each cell fraction was analyzed by flow cytometry to confirm enrichment of the desired cell subtype. It is important to note that this method does not eliminate all other cell types.

## Flow Cytometry labeling for analysis of isolated cells

Two million cells were mixed with 3.0 µl of Fc block and 0.25 µl of Zombie NIR in a volume of 200 µl of staining buffer (MACS staining buffer + 5% BSA) and incubated at 4 °C for 15 min. After 15 min we added 2 ml of staining buffer and centrifuge at $491 \times g$ for 5 min. Cell pellets were resuspended in 200 µl of staining buffer. Samples were then incubated with antibodies at 4 °C for 30 min in darkness. After incubation, 2Ml of staining buffer was added, and cells were centrifuged at $491 \times g$ for 5 min. Cells were washed 2x with staining buffer and resuspended in 300 µl of staining buffer for flow cytometry. Antibodies for identification panel: anti-CD45 (clone: 30-F11, Fluorochrome: BV421, Cat#: 103134, Manufacturer: Biolegend, titer per 100 µl staining volume: 1 µl), anti-CD11b (clone: M1/70, Fluorochrome: BV605, Cat#:101257, Manufacturer: Biolegend, titer per 100 µl staining volume: 1 µl), anti-CD11c (clone: N418, Fluorochrome: PE/Cy7, Cat#: 117318, Manufacturer: Biolegend, titer per 100 µl staining volume: 1 µl), anti-CD8a (clone: 53-6.7, Fluorochrome: BV510, Cat#: 100752, Manufacturer: Biolegend, titer per 100 µl staining volume: 1 µl), anti-CD4 (clone: GK1.5, Fluorochrome: BV785, Cat#: 100453, Manufacturer: Biolegend, titer per 100 µl staining volume: 1 µl), anti-CD25 (clone: PC61, Fluorochrome: PE, Cat#: 102008, Manufacturer: Biolegend, titer per 100 µl staining volume: 1 µl), anti-CD25 (clone: PC61, Fluorochrome: PE/Cy5, Cat#: 102010, Manufacturer: Biolegend, titer per 100 µl staining volume: 1 µl), anti-CD19 (clone: 6D5, Fluorochrome: PE/Cy7, Cat#: 115520, Manufacturer: Biolegend, titer per 100 µl staining volume: 1 µl), anti-I-A/I-E (clone: M5/114.15.2, Fluorochrome: PerCP/Cy5.5, Cat#: 107626, Manufacturer: Biolegend, titer per 100 µl staining volume: 1 µl).

FMO controls were prepared for the respective markers.

## In vivo biopanning

For in vivo biopanning, we intravenously injected $1.5 \times 10^{12}$ phages in 150 µl of PBS pH7.2 into Py8119 and Py117 tumor-bearing CL57BL/6 mice. Tumor and LN samples were processed for immune cell isolation 2 h post injection of the phage library. Three tumor-bearing animals were used for every tumor type in each round of biopanning. For the preliminary round of CD45⁺ cell-specific phage selection (BP0), we isolated immune cells from tumor tissue as described above, and selectively isolated CD45⁺ cells by using CD45⁺-positive selection magnetic beads after dead cell removal. Phage particles enriched in the preliminary round of biopanning were amplified and prepared for BP1. For BP 1 through 4, we isolated immune cells following ficoll gradient protocol, performed dead cell removal, then selectively isolated six different cell types: (Total TIL (CD45⁺), CD11b⁺, CD11c⁺CD11b⁻, CD8⁺, CD4⁺CD25⁻, and CD4⁺CD25⁺). We used the phage library amplified from the total TIL population for subsequent rounds of biopanning.

## Amplification of M13 phages

We produced the M13 phage library by transforming 2 µg of recombinant phagemid DNA in TG1 electrocompetent *E.coli* cells and by adding M13KO7 helper phage to a final titer of $10^9$/ml. We used ampicillin at a final concentration of 50 µg/ml for selection with phage particles then precipitated using 0.3 volume of PEG8000 after centrifuging out the bacterial pellet. Phages were titered by serially diluting the phage library and plating them after infecting them with TG1 logarithmic culture for 30 min at 37 °C. For the amplification of phages in between biopanning, 50 µl of ficoll-separated immune cells or CD45⁺-sorted cells were mixed with 200 µl of logarithmic TG1 bacterial culture and incubated at 37 °C for a minimum of 30 min. Next, 1 ml of 2-YT medium supplemented with 50 µg/ml of ampicillin was added. M13KO7 helper phage was added to a final titer of $10^9$/ml, and the phage-infected bacterial culture was transferred to sterile 250 ml 2-YT broth with 50 µg/ml of ampicillin and incubated overnight at 37 °C with shaking at 200 RPM. To harvest phage particles, we centrifuged bacterial culture at $4415 \times g$ for 10 min at 4 °C. The culture-free medium was precipitated with 0.3 volume of PEG8000 and incubated at 4 °C for 2 h. Phage precipitate was concentrated by centrifugation at $4415 \times g$ for 20 min at 4 °C. The phage pellet was suspended in 5 ml of sterile PBS pH 7.2 and filtered through 0.22 µM low-protein binding filter. Phages were titered as previously described.

## Selection of phage clones and Sanger sequencing

Phage clones with nanobody inserts were screened by colony-PCR during phage titering of immune cells (CD45⁺, CD11b⁺, CD11c⁺CD11b⁻, CD8⁺, CD4⁺CD25⁻, CD4⁺CD25⁺) isolated from biopanning 3 and 4. Phage clones were selected from plates of higher dilutions to target highly enriched clones. Nanobody-positive clones were inoculated in 2-YT medium supplemented with 50 µg/ml of ampicillin and recombinant phagemid DNA extracted after overnight incubation at 37 °C. Phagemid-extracted DNA was labeled and stored at −20 °C for future use. Phage clones were Sanger sequenced at the UT Southwestern Medical Center DNA sequencing core facility using phiS2 5′-ATGAAAT ACCTATTGCCTACGG forward and psiR2 5′-CGTTAGTAAATGAAT TTTCTGTATGAGG reverse primer sequences.

## Next-generation sequencing (NGS)

For NGS sample preparation, 25 µl of each immune cell subtype isolated either by MACS magnetic beads or flow cytometry sorting were mixed with 200 µl of logarithmic TG1 bacterial culture and incubated for 2 h at 37 °C. After incubation, phagemid DNA was extracted from bacterial culture using the QIAprep spin mini prep plasmid extraction kit (Qiagen) with OD260 value was measured by NanoDrop™2000 (ThermoFisher, NY). Phagemid DNA was amplified for five cycles using the phiS2 forward and psiR2 reverse primers. The amplicons were further amplified for 20 cycles and the resulting amplicons were resolved in 1.5% agarose gel to cut out band sizes ranging from 550 to 850 bp. Amplicons were concentrated by QIAquick gel extraction kit and estimated by Bioanalyzer (Agilent). The NGS sequence library was

created by ligating illumina universal sequence adapter with the bar-code for MISeq paired 300 bp reads (Illumina). NGS sequencing was performed at the UT Southwestern Medical Center NGS core facility.

## NGS data analysis

NGS analysis began with removal of remaining adapters and primers that carried over from PCR amplification and sequencing. Poor quality reads were subsequently removed. To construct the VHH antibodies from NGS sequencing, we first used MiXCR V3 to align the reads to the IMGT Alpaca IG reference library with default settings. Then, we fully assembled all regions by changing the assembling features to include not just CDR3, but all regions from FR1 to FR4, including CDR1 and CDR2. This method ensures VHHs' full reconstruction of the VHHs. For any further analysis, we exported the fully assembled VHHs, removed partially assembled VHHs to VDJtools and/or Immunarch for explora-tory visualization, diversity calculation, overlapping, antibody tracking and enrichment[45,46]. Diversity was calculated via commonly used indices such as Simpson and Shannon. True diversity or the effective number of types was defined as the number of equally abundant types needed for the average proportional abundance of the types to equal that observed in the dataset of interest, where all types may not be equally abundant[47].

## Expression and purification of nanobody

We subcloned each nanobody clone sequence in the pET21d+ vector between NheI and XhoI sites. Subcloned nanobody-expressing vectors were transformed in BL21 (DE3)-pLysS-competent cells and a single colony was cultured in 50 ml of 2YT broth supplemented with 50 µg/ml ampicillin in an orbital shaker at 37 °C with rotation at 200 RPM. After overnight incubation, the culture was transferred to 500 ml of 2-YT broth in a 1 L conical flask and incubated further until the OD600 reached ~0.6. Induction was then performed using IPTG at a final concentration of 500 µM. Bacterial cells were pelleted by cen-trifugation at 4816 × $g$ for 10 min at 4 °C. The culture pellet was sus-pended with 5 ml of bacterial lysis buffer (10 mM imidazole, 300 mM NaCl, 0.1 % Triton X-100 in phosphate buffer pH 8.0) freshly supple-mented with lysozyme and kept in ice for 30 min to ensure bacterial disruption, then sonicated three times with a 5 s on, 5 s off cycle at 60% amplitude. The lysate was centrifuged at 4816 × $g$ for 20 min at 4 °C, and the supernatant was transferred to a 15 ml conical tube for ÄKTA pure purification. His-tag purification was performed by using 1 ml HisTrap HP column in ÄKTA pure 25 (Cytiva, MA). The protocol included binding for 5 min in the presence of phosphate buffer with 300 mM NaCl and 10 mM imidazole, washing for 15 column volumes with a flow rate of 1 ml/min in the presence of phosphate buffer with 300 mM NaCl and 20 mM imidazole. Finally, a linear gradient of imi-dazole ranging from 50 to 300 mM was used for elution. Fractions were resolved in 15% SDS-PAGE and the pure fractions were pooled and concentrated by passing through 30 kDa and 3 kDa cut off Amicon centrifugal filter unit. Protein estimation was performed by nanodrop OD280 and BCA assay (ThermoFisher, NY). Nanobody proteins were confirmed by immunoblotting using anti-His-tag (clone: polyclonal, Cat#: 2365S, Manufacturer:Cell Signaling Technology 1 µl for 1000 µl blocking buffer) and anti-VHH (clone: 96A3F5, Cat#: A01860, Manu-facturer: GenScript, 1 µl for 1000 µl blocking buffer) antibodies.

## Protein binding by flow cytometry

Target binding of nanobody proteins was assessed by flow cytometry analysis. Immune cells from naïve spleen, tumor, draining lymph nodes, and spleen from Py8119 or Py117 tumor-bearing mice as described previously were suspended in staining buffer. Two million cells were mixed with 3.0 µl of Fc block and 0.25 µl of Zombie NIR, incubated at 4 °C for 15 min. After 15 min we added 2 ml of staining buffer and centrifuged at 491 × $g$ for 5 min. For His-labeled nanobody alone, protein (2–4 µg) was added to the Fc block-treated cells with a

final reaction volume of 100 µl and incubated for 30 min at 4 °C. After washing once with staining buffer, anti-His-tag (clone: J095G46, Fluorochrome: APC, Cat#: 362605, Manufacturer: Biolegend, titer per 100 µl staining volume: 1 µl) was added, and the mixture was incubated further for 15 min at 4 °C. For a negative control, APC-His-tag antibody was added to the cells treated with no nanobody proteins. For Venus fusion proteins 2–4 µg were incubated with no secondary antibodies to then detect protein directly using venus fluorescence. Cells were washed 2× with staining buffer and incubated with antibodies for immune cell identification at 4 °C for 30 min in darkness and washed 2× with staining buffer after incubation. Cells were suspended in 300 µl of staining buffer for flow cytometry. The following panel of antibodies was used to analyze nanobody binding in specific subtypes of immune cells: anti-CD45 (clone: 30-F11, Fluorochrome: BV421, Cat#: 103134, Manufacturer: Biolegend, titer per 100 µl staining volume: 1 µl), anti-CD11b (clone: M1/70, Fluorochrome: BV605, Cat#: 101257, Manu-facturer: Biolegend, titer per 100 µl staining volume: 1 µl), anti-CD11c (clone: N418, Fluorochrome: AF488, Cat#: 117311, Manufacturer: Bio-legend, titer per 100 µl staining volume: 1 µl), anti-CD8a (clone: 53-6.7, Fluorochrome: BV510, Cat#: 100752, Manufacturer: Biolegend, titer per 100 µl staining volume: 1 µl), anti-CD4 (clone: GK1.5, Fluor-ochrome: BV785, Cat#: 100453, Manufacturer: Biolegend, titer per 100 µl staining volume: 1 µl),, anti-CD25 (clone: PC61, Fluorochrome: PE/Cy5, Cat#: 102010, Manufacturer: Biolegend, titer per 100 µl staining volume: 1 µl), anti-CD19 (clone: 6D5, Fluorochrome: PE/Cy7, Cat#: 115520, Manufacturer: Biolegend, titer per 100 µl staining volume: 1 µl), anti- I-A/I-E (clone: M5/114.15.2, Fluorochrome: PerCP/Cy5.5, Cat#: 107626, Manufacturer: Biolegend, titer per 100 µl staining volume: 1 µl). Ultracomp eBeads (ThermoFisher) were used for com-pensation. Cells were analyzed on the LSR Fortessa (BD Biosciences).

## Immunoprecipitation of nanobody targets and mass spectrometry

To identify nanobodies specific target proteins, immunoprecipita-tion (IP) analysis was performed using membrane protein extracts of mouse splenocytes. Membrane protein was isolated with Mem-PER™ plus membrane protein extraction kit (ThermoFisher). For IP, Protein G Dynabead (ThermoFisher) was washed with PBS, mixed with 5 µl of anti-his-tag antibody, incubated for 1 h at RT, and washed three times with PBS. Nanobody protein at a concentration of 10 µg was mixed with 500 µg of membrane protein in PBS and mixed with Dynabead-antibody complex after 1 h incubation at 4 °C. The nanobody-membrane protein mixture was incubated further for 2 h at 4 °C, washed three times with PBS, pelleted, and resuspended in 15 µl of water with 15 µl of 4× protein loading dye (BioRad, Hercules, CA). The sample was resolved in 4 to 15% gradient acrylamide gel and stained with silver stain (BioRad, Hercules, CA). Alternatively, nanobody-IgG-Fc fusion protein was used in place of anti-his-tag antibody.

Samples were digested overnight with trypsin (Pierce) following reduction and alkylation with DTT and iodoacetamide (Sigma–Aldrich). The samples then underwent solid-phase extraction cleanup with an Oasis HLB plate (Waters) and the resulting samples were injected onto a QExactive HF mass spectrometer coupled to an Ultimate 3000 RSLC-Nano liquid chromatography system. Samples were injected onto a 75 µm i.d., 15-cm long EasySpray column (Thermo) and eluted with a gradient from 0 to 28% buffer B over 90 min with a flow rate of 250 nL/min. Buffer A contained 2% (v/v) ACN and 0.1% formic acid in water, and buffer B contained 80% (v/v) ACN, 10% (v/v) trifluoroethanol, and 0.1% formic acid in water. The mass spectrometer operated in positive ion mode with a source voltage of 2.2 kV and an ion transfer tube temperature of 275 °C. MS scans were acquired at 120,000 resolutions in the Orbitrap and up to 20 MS/MS spectra were obtained for each full spectrum acquired using higher-energy collisional dissociation (HCD) for ions with charges 2–8.

Dynamic exclusion was set for 20 s after an ion was selected for fragmentation.

Raw MS data files were analyzed using Proteome Discoverer v2.4 SP1 (Thermo), with peptide identification performed using Sequest HT searching against the mouse reviewed protein database from UniProt. Fragment and precursor tolerances of 10 ppm and 0.02 Da were specified, and three missed cleavages were allowed. Carbamidomethylation of Cys was set as a fixed modification and oxidation of Met was set as a variable modification. The false-discovery rate (FDR) cutoff was 1% for all peptides. Mass spectrometry experiments were conducted at the Proteomics Core Facility, UT Southwestern Medical Center, Dallas, Texas.

## Single-cell RNA sequencing

In preparation for the scRNAseq experiment three mice bearing Py8119 tumors were injected with CD45+-specific phage recovered from BP1 and CD45+ cells and were sorted from the digested tumor (as per "Isolation of immune cells" section) using the AriaII flow sorter (BD Biosciences). Phage were recovered from sorted cells, amplified, and injected into three more mice for round two. Upon digestion of the tumor in round two, a fraction of CD45+ cells were frozen in 90% FetalClone II and 10% DMSO after magnetic microbead positive selection. The remainder of the digested cells were sorted for CD8+CD11b-CD11c-SSClo (CD8 T cells) and CD11c+MHCII+CD11b- (DCs). Phage were recovered from sorted cells, amplified, and each of the CD8 T cell- and DC-specific phage pools were injected into three more tumor-bearing mice. After 2 h, tissue was mechanically minced and digested for 40 min with 100 U/ml Collagenase IV (Worthington), 10 mg/ml DNAse (Sigma Aldrich) and 10.5 mM Y-27632 (Sigma Aldrich) in HBSS (Corning) and RPMI 1640 (Mediatech, Inc., Corning) complemented with 3% FetalClone II (HyClone). Immune cells were enriched using CD45 magnetic microbeads. Viability and cells count was measured using the 0.4% Trypan blue exclusion method. Cells were then frozen as above. For controls, three additional tumor-bearing mice were injected with PBS (PBS, Fig. 4) and three more with phage without a nanobody (Phage, Fig. 4). Two hours later, tumors were digested with the collagenase IV protocol and the CD45+ fraction was collected as above. At the time of sequencing, thawed cells were counted using a TC20 automated cell counter (BioRad) and adjusted to 1000 cells/µl in 0.04% BSA/PBS and 1 ml/sample were submitted to the Next Generation Sequencing Core, McDermott Center, UT Southwestern Medical Center. Cells were loaded according to standard protocol of the Chromium single cell 3' kit, capturing 10,000 cells (V3 chemistry). All subsequent procedures, including library construction, were performed according to the standard manufacturer's protocol. Single-cell libraries were sequenced on NovaSeq (Illumina) to a depth of 50,000 reads per cell.

## Single-cell RNA-seq data processing

Raw data and FASTQs were generated by the Next Generation Sequencing Core at UT Southwestern. Using Cell Ranger version 5.0.1 pipeline, FASTQ files were aligned to the mouse reference genome (mm10) using STAR version 2.7.2a and counted using Cell Ranger count with default parameters and recommendations. The counting matrix was imported into Seurat (3.2.3) via R (4.0.3) for quality assessment and downstream analysis. Cells were filtered by excluding cells with less than 200 genes, all genes in less than three cells, and genes expressed as being composed of greater than 20% mitochondrial genes. Data were then normalized using the NormalizeData function with default parameters. Variable genes were detected using the FindVariableFeatures function. Cell cycle scores were calculated by the CellCycleScoring function, and a cell cycle difference was calculated by subtracting the S phase score from the G2M score. Data were scaled and centered using linear regression on the counts and the cell cycle score difference. Principle component analysis was run with the RunPCA function using default parameters. Batch effects were corrected, and samples were integrated by matching mutual nearest neighbors (fastMNN) using the Seurat-Wrappers library with default parameters. Cell clusters were identified via the FindNeighbors and FindClusters functions, with 0.5 resolution and UMAP clustering algorithms. A FindAllMarkers table was created, and clusters were defined by using SingleR and celldex with ImmGenData from (https://www.immgen.org/)[48], and finally with canonical markers (Supplementary Fig. 7c). Gene Ontology (GO) enrichment for each cluster and between samples was performed using topGO version 2.42.0 and differentially expressed genes. GO annotations were obtained from the Bioconductor database org.Mm.eg.db version 3.12.0. Graphs and plots were generated by dittoseq version 1.2.5, SCP version 0.2.6 or Seurat.

## Colocalization with PHB2

MC38 colorectal cancer cells or H1299 human non-small cell lung carcinoma were seeded on a 22 x 22 mm sterile coverslip and grown for 48 h in DMEM + 10 % Fetal Clone II or 24 h in RPMI + 10% Fetal Clone II respectively. Cells were fixed with 4% paraformaldehyde in PBS for 20 min at 4 °C. Coverslips were then washed in PBS and blocked with Cell Staining Buffer (Biolegend) for 40 min at 4 °C, permeabilized (0.1% TritonX100, 0.025% sodium azide in PBS) for 30 min at 4 °C and further blocked with Cell Staining Buffer for 20 min at 4 °C. Cells were then stained with mouse anti-PHB2 antibodies (clone: 1D9C7, Cat#: 50 173 6851, Manufacturer: Thermo Fisher Scientific, titer per staining volume: 1 µl) and Nb1 (2 µg) in 200 µl Cell Staining Buffer for 30 min at 4 °C. Cells were subsequently washed with PBS and stained with rabbit anti-camelid VHH antibodies (clone: 96A3F5, Cat#: A01860, Manufacturer: Genscript, titer per staining volume: 2 µl) in 200 µl Cell Staining buffer for 30 min at 4 °C. Following anti-VHH antibody, cells were washed and stained with goat anti-mouse (clone: polyclonal, Fluorochrome: Alexa Fluor 488, Cat#: A11029, Manufacturer: Thermo Fisher Scientific, titer per staining volume: 1 µl) and goat anti-rabbit antibodies (clone: polyclonal, Fluorochrome: APC, Cat#: A10931, Manufacturer: Thermo Fisher Scientific, titer per staining volume: 1 µl) in 200 µl Cell Staining buffer for 90 min at 4 °C. Coverslips were counterstained with 20 nM Hoechst 33342 for 5 min in Cell Staining Buffer (Thermo Fisher Scientific) then mounted on slides with Vectashield Antifade Mounting Medium (Vector laboratories). Control samples were stained with anti-PHB2 and anti-VHH antibodies or with anti-VHH alone. All controls were stained with goat anti-mouse-Alexa Fluor 488 and goat anti-rabbit-APC antibodies

Images were acquired on a Nikon CSU-W1 SoRa inverted spinning disk confocal microscope at the UT Southwestern Quantitative Light microscopy Core Facility (Thanks to NIH 1S10OD028630-01 grant awarded to Dr. Kate Luby-Phelps). Using a PlanApoλ 100× objective, 20 fields of view were imaged with 0.2 µm steps in Z-orientation to capture the full cell outlines. The laser and exposure parameters ensured a signal to noise ratio > 3:1. (1) DAPI channel at 405 nm laser (10% power) with 50 ms exposure. (2) AF488 channel at 488 nm laser (10% power) with 50 ms exposure. (3) APC channel at 640 nm laser (1% power) with 20 ms exposure. Analysis of images was performed using Fiji ImageJ version 2.3.051. For colocalization Fiji plugin JaCOP version 2.1.4 was used[49]. Mander's colocalization coefficient M1 describes the contribution of pixels from image A (Nanobody signal) colocalizing on pixels of image B (PHB2 signal)[49,50]. For analysis, thresholds were adjusted to exclude background noise, as previously described[49]. To test significance of colocalization, M1 was determined after Image A was translated as a geometric transformation on X and Y axes in ImageJ (XY pixel shift), with p-values derived using Student's T-test[51]. For figures, thresholds were set between 100 (mean background intensity) and 500 for all 16-bit images.

## Modeling of Nb1 binding with PHB2

RosettaSuite version 3.13 was used in this project. The software is freely available for academia, along with documentation, at the address: www.rosettacommons.org.

Structural models of Nb1 were generated and scored with the Rosetta Antibody framework starting from its amino acid sequence[31,32]. Briefly, regions of high sequence identity such as the framework region, CDR H1 and CDR H2, were modeled based on homologous templates using the antibody application, while the highly variable CDR H3 was reconstructed de novo via loop modeling using the antibody_H3 application[33]. A total of 10,000 models were generated and then clustered based on their structure similarity using energy_based_clustering[34]. The lowest score model for the best eight clusters were then selected for docking.

The structural model for the full length PHB2 was obtained from the AlphaFold Database, searching for "Prohibitin-2" and "mouse"[36]. Prior to docking, the predicted structure of PHB2 was refined with Rosetta using the movers PackRotamerMover and MinMover were used to repack the sidechains and minimize the backbone, and FastRelax to relax the overall structure. The mover AlignChain aligned the relaxed structure to the native one, while the RMSD metric was used to calculate the root-mean square deviation. To decrease computational time, the intracellular and transmembrane regions (aa 1–38) were removed using DeleteRegionMover.

The global docking protocol within Rosetta was then performed using the truncated PHB2 and the top eight NB1 models in parallel[32,37,38]. Eight input pdb files, containing both PHB2 and a Nb1 variant, were prepared using the AddChain mover. The chains were renumbered using the clean_pdb.py script. To optimize the computational time and to ensure that the nanobody was correctly oriented towards the target, a constraint was set to have at least one CDR residue on Nb1 in contact with PHB2. AmbiguousConstraints were set for each of the CDR residue on Nb1 as follow:

SiteConstraint CA <residue> A FLAT_HARMONIC 0 1 10

For global docking, the following protocols have been set up in a single instruction file: ConstrainSetMover to set the constraints, DockSetupMover and DockingInitialPerturbation to test random initial orientations of PHB2 and Nb1, Docking (low resolution), SaveAndRetreiveSidechains and Docking (high resolution) for standard docking, ClearConstraintsMover to remove the constraints, FastRelax to minimize the overall complex and InterfaceAnalyzerMover to calculate binding energy and identify the binding interface. The docked models were then analyzed using the residue_energy_breakdown application and clustered based on conformational similarity using energy_based_clustering[34].

The 11 top binding sites identified through global docking were then re-docked using SnugDock, a local docking protocol which allows flexibility in the CDR regions of the nanobody[39,40,52]. To prepare the files according to the SnugDock format, each of the 11 initial Nb1-PHB2 complexes was processed using SwitchChainOrder and PyIgClassify.py. The latter is also available online at: http://dunbrack2.fccc.edu/PyIgClassify/User/UserPdb.aspx. The SnugDock docking was performed using the snugdock application within Rosetta, and 1000 models were generated for each of the 11 PHB2-Nb1 complexes. Only binding sites 1, 5, 6 and 8 resulted more favorably in terms of binding strength of binding and the remaining seven sites were discarded. To inspect nanobody binding when PHB2 is in an hypothetical multimeric state, the crystal structure of the coiled coil domain of PHB2 (PDB ID: 6IQE) was used to generate a pentameric version of the protein, named PHB2 5x_CC[35]. The pentamer was refined within Rosetta as described above for the full length PHB2, with the addition of a step to calculate binding energy between one monomer and the remaining four after relaxation (using InterfaceAnalyzerMover). The best docking complexes of Nb1 and PHB2 from SnugDock were aligned to the central coil in 5x_CC using Pymol, and the resulting structures were visually inspected for clashes between Nb1 and the additional coiled coils.

## Statistics and reproducibility

The study was designed to have at least 3 replicate mice for each round of biopanning. There were no data excluded from reporting. There was random group assignment for all mice whose tumors were in the appropriate size range. There were additional samples collected for the scRNAseq sorting experiment, but those data will be presented in a different report. The investigators were not blinded to allocation during experiments and outcome assessment.

## Reporting summary

Further information on research design is available in the Nature Portfolio Reporting Summary linked to this article.

## Data availability

Materials, protocols, and data are available upon request. Some requests may be subject to materials transfer agreement (MTA) (Lead Contact: Todd.Aguilera@utsouthwestern.edu). Specific sequences and enrichment information will be made available after an executed data use agreement and phage libraries would be made available with a materials transfer agreement. This will be made available for scientific research purposes, reproducing data, and to build on the findings. There will be restrictions on public disclosure of specific sequences and commercial development. All other materials and protocols are available upon request. Phage sequencing data is stored on the UTSW Radiation Oncology TrueNAS server behind UTSW firewall that can be transferred upon execution of appropriate agreements. Single cell RNA sequencing data has been deposited at GEO, accession number #GSE223428. Mass Spectrometry data for Nb1 target identification has been deposited on PRIDE, accession number # PXD046363. Source data are provided with this paper.

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

## Acknowledgements

This work was supported by the CPRIT First-Time Tenure-Track Recruitment award RR170051 (T.A.A.), the Presidents Research Council Distinguished Researcher Award (T.A.A.), the Carroll Shelby Family Foundation (T.A.A.), Cancer Biology Training Grant T32 CA 124334 (K.S.), and the Mary Kay Ash Charitable Foundation International Postdoctoral Scholar Fellowship (S.D.). We thank the Moody Foundation Flow Cytometry Core at Children's Research Institute, the McDermott Center Next Generation Sequencing Core, the Proteomics Core facility and the Quantitative Light Microscopy Core at UT Southwestern and the Center for Structural Biology and the Advanced Computing Center for Research and Education at Vanderbilt University.

## Author contributions

Conceptualization, T.V.S., E.A.E., M.W. and T.A.A.; Methodology, T.V.S., E.A.E., J.M., C.E.M., M.W., and T.A.A.; Software, E.A.E., and C.E.M.; Validation, T.V.S., E.A.E., K.L.S., S.D., I.G., P.Q.L., C.H., C.E.M., and T.A.A.;

Formal Analysis; T.V.S., E.A.E., and C.E.M.; Investigation, T.V.S., E.A.E., K.L.S., S.D., I.G., P.Q.L., C.H., C.E.M., M.W., and T.A.A.; Resources; T.A.A. and J.M.; Data Curation, T.V.S., E.A.E., S.D., and C.E.M.; Writing-Original Draft, T.A.A. T.V.S., and E.A.E.; Writing- Reviewing Editing; T.V.S., E.A.E., S.D., C.E.M., and T.A.A.; Visualization; T.V.S, E.A.E, C.E.M., S.D., and T.A.A., Supervision; T.A.A., J.M., and M.W.; Project Administration; T.A.A.; Funding Acquisition, T.A.A., and J.M.

## Competing interests

The authors declare no competing interests.
