## [Peer Review File · Nature Communications]

Simultaneous selection of nanobodies for accessible epitopes on immune cells in the tumor microenvironmentREVIEWER COMMENTS

Reviewer #1 (Remarks to the Author):

Sekar et al reported an in vivo platform that can isolate the enriched nanobodies binding to various immune cells in the tumor microenvironment, and they named the technology INSPIRE-seq. Technically, they injected the naïve phage display nanobodies in phage form to syngenic mice with transplanted breast cancer (immune sensitive and immune resistant tumors), followed by isolation of various types of the immune cells and identification of the cell associated phage displayed VHHs by NGS sequencing. They demonstrated that they could isolate multiple VHHs for each type of immune cells from tumors and also draining lymph nodes, and observed the enriched VHH clones in repeated bio-panning. Interestingly they also described that in parallel enrichment studies they could identify the correlation of the enriched VHHs and the lineage/transcriptome associated changes in single cells via scRNA-seq analysis (no extensive analysis presented on the detailed correlation, though as presented).

As an example, the authors showed that one nanobody enriched in and isolated from mouse dendritic cells, and further demonstrated, by IP coupled with mass spectrometry analysis, PHB2, a cell multi-transmembrane domain-containing protein. However, the detailed role of PHB2 in regulating the immune function and tumor immunology were not very clear. Various robust experiments were performed to show that the VHH specifically bound to the cell surface PHB2. They further sought to model the potential binding topology of the VHH to PHB2, using the interesting alfa-fold AI structure-based prediction of the VHH and PHB2 interaction.

Overall, the studies established an interesting platform that can screen and identify the VHHs that are enriched in various types of immune cells or even stromal cells, via unbiased approach. The correlation of the VHH1 binding and the single cell transcriptome changes identified by the parallel scRNA-seq analysis should be also informative and fruitful. It is conceivable that the identification of the VHH binders to specific types of immune cells, or even stromal cells, and coupled with the identification of the VHH-binding association of potential transcriptome/phenotype changes of the bound cells, should be quite revealing. While it is also interesting to further identify the target of one VHH that preferentially bound to the dendritic cells, PHB2, the studies fell short, in this reviewer's opinion, to significantly demonstrate the particular value of the in vivo screening to have the capacity to identify the key target that can modulate the immune cell anti-tumor activity to improve cancer immune therapy. This lack of evidence reduces the enthusiasm of this reviewer. Below are the detailed comments:

1. The title did not seem to accurately describe the experiments and the results presented by the paper, and should be improved.
2. Figure 1. No detailed or traceable description/reference/methods were described for construction of the naïve llama phage display library. As this work formed the foundation of the studies, it is essential to be included with adequate details.
3. While the authors claimed that the binding of a particular VHH to the single cells can correlate to the

RNA expression changes in the TME immune cells, the adequate data was not shown in details in the manuscript. It would be more convincing to thoroughly present the data, if available.

4. It is intriguing to find Nb1 antigen target, PHB2, and impressive to identify the antigen and convincingly demonstrate the binding of Nb1 to PHB2. However, it is not clear whether binding of the immune cells by Nb1 affects the dendritic function in antigen presentation or immune cell activation.

5. It is also important to demonstrate whether the INSPIRE method can isolate a VHH that not only targeting an immune cell in TME, but also modulating the function of the immune cell via the binding to improve the anti-cancer immunity. This is a crucial point, because lacking of this does not distinguish the unique benefit of using INSPIRE-seq in identifying the VHH binders to the immune cells in TME in vivo, as the simple VHH phage panning ex vivo using the isolated TME immune cells ex vivo is supposed to do the similar or same thing, substantially reducing the impact of the technology.

Reviewer #2 (Remarks to the Author):

In this paper, the authors developed INSPIRE-seq to select immune cell-binding nanobodies that penetrate the tumor microenvironment. It may be a good strategy for drug discovery in the future. However, we still have several concerns on this paper:

1. There's a series of writing mistakes in this manuscript. For example, "to for" in the first line of the abstract. The authors should check over the whole manuscript to improve the readability.
2. The authors described "A camelid based VHH nanobody bacteriophage library was derived from peripheral blood mononuclear cells of sixteen non-immunized llamas". However, the immune background seems not included in the manuscript. It should be detailed that how they get the important materials in such a biopanning pipeline.
3. Since that the biopanning was performed in two mouse model, the cross-species binding ability should be further confirmed. Could it bind to the homologous protein on the surface of human immune cells?
4. The authors declared that "greater DC activation pathway enrichment in the CD8 and CD11c samples". However, genes in Figure 4I seem to be the markers of DC or T cell subsets, which is not surprising to be found that higher in specific cell types. Our concern is that this conclusion is not invalid.
5. The authors identified a nanobody that binds to PHB2 on the surface of cDC1. However, they did not evaluate the downstream effect of this binding process. Could it activate the maturation of cDC1 cells? Could it promote the anti-tumor effect of CD8+ T cells? If not, what is the translational value of this nanobody?

Reviewer #3 (Remarks to the Author):

The manuscript, "Simultaneous selection of nanobodies for immune epitopes in the complex tumor microenvironment" describes a new method called INSPIRE-Seq to select for nanobodies in vivo using next generation sequencing. The paper aims to build upon existing in vivo phage display technologies by sorting immune cells in the tumor microenvironment to identify nanobodies enriched in different subpopulations. This work is interesting and is one of few papers that uses an unbiased approach to identify nanobodies selective for certain immune subpopulations in different organ sites. Publication should be considered, although significant issues must be addressed prior:

1. Py8819 and Py117 are used as models of differing immune response to therapy like radiation in the introduction. Throughout the rest of the study, these two tumors are used for biopanning of nanobodies, but there is no indication/discussion of differences between nanobodies in the figures or text. This should be addressed
2. In Figure 1A as well as other figures, S nozzle N cartoon are mentioned, but they are not explained.
3. Additional evidence is needed to determine if the IgGFc-Nb1 binds to Prohibitin-2. Based on the table in supplementary Figure 6B, most of the abundant proteins are intracellular so unlikely to be the binding target, but DNAJC9 has been shown to interact with PHB2 (Bavelloni A. et al. IUBMBC 2015). IgGFc-Nb1 could be binding to DNAJC9, and the confocal microscopy and immunoprecipitation/western blotting demonstrate IgGFc-Nb1 is binding to DNAJC9 interacting with PHB2. Confocal microscopy of IgGFc-Nb1 and DNAJC9 would be sufficient to show that IgGFc-Nb1 does not colocalize with DNAJC9. If it does colocalize with DNAJC9, further experiments using blocking by a DNAJC9-specific antibody and PHB2-specific antibody are needed to show antigen specificity.

Reviewer #4 (Remarks to the Author):

I was asked to evaluate the proteomics portion of this manuscript. Overall, this experiment appears to be well-designed. However, it is not possible to evaluate this work because the authors did not provide the data and experimental details that are necessary to do so. The authors should upload all LC-MS/MS data files, experimental details, and database search results to a public repository such as MassIVE (at UCSD). This is a standard requirement for publication of proteomics data.

Nature Communications Reviews:
Response to reviewer comments:

Reviewer #1 (Remarks to the Author):

Sekar et al reported an in vivo platform that can isolate the enriched nanobodies binding to various immune cells in the tumor microenvironment, and they named the technology INSPIRE-seq. Technically, they injected the naïve phage display nanobodies in phage form to syngenic mice with transplanted breast cancer (immune sensitive and immune resistant tumors), followed by isolation of various types of the immune cells and identification of the cell associated phage displayed VHHs by NGS sequencing. They demonstrated that they could isolate multiple VHHs for each type of immune cells from tumors and also draining lymph nodes, and observed the enriched VHH clones in repeated bio-panning. Interestingly they also described that in parallel enrichment studies they could identify the correlation of the enriched VHHs and the lineage/transcriptome associated changes in single cells via scRNA-seq analysis (no extensive analysis presented on the detailed correlation, though as presented).

As an example, the authors showed that one nanobody enriched in and isolated from mouse dendritic cells, and further demonstrated, by IP coupled with mass spectrometry analysis, PHB2, a cell multi-transmembrane domain-containing protein. However, the detailed role of PHB2 in regulating the immune function and tumor immunology were not very clear. Various robust experiments were performed to show that the VHH specifically bound to the cell surface PHB2. They further sought to model the potential binding topology of the VHH to PHB2, using the interesting alfa-fold AI structure-based prediction of the VHH and PHB2 interaction.

Overall, the studies established an interesting platform that can screen and identify the VHHs that are enriched in various types of immune cells or even stromal cells, via unbiased approach. The correlation of the VHH1 binding and the single cell transcriptome changes identified by the parallel scRNA-seq analysis should be also informative and fruitful. It is conceivable that the identification of the VHH binders to specific types of immune cells, or even stromal cells, and coupled with the identification of the VHH-binding association of potential transcriptome/phenotype changes of the bound cells, should be quite revealing. While it is also interesting to further identify the target of one VHH that preferentially bound to the dendritic cells, PHB2, the studies fell short, in this reviewer's opinion, to significantly demonstrate the particular value of the in vivo screening to have the capacity to identify the key target that can modulate the immune cell anti-tumor activity to improve cancer immune therapy. This lack of evidence reduces the enthusiasm of this reviewer. Below are the detailed comments:

We thank the reviewer for their insight and enthusiasm, yet we recognize their concern that we have not identified “the key target that can modulate the immune cell anti-tumor activity to improve cancer immune therapy.” Although this is our goal hopefully the subsequent responses will clarify the scope of this manuscript and that the reviewers will recognize these findings are critical and important to achieve our long-term goals.

1. The title did not seem to accurately describe the experiments and the results presented by the paper, and should be improved.

Thank you for this comment. We have considered this statement carefully and made a small change to describe the experiments and results more accurately.

“Simultaneous selection of nanobodies for accessible epitopes on immune cells in the tumor microenvironment”

Justification of the title:

- **‘Simultaneous selection of nanobodies’:** We show for the first time a method for parallel enrichment of nanobodies for 5 different immune cell subtypes using in vivo phage display. Therefore, in one round as well as at the end of the biopanning rounds we get nanobody enrichment and selectivity information for all 5 cell subtypes simultaneously.
- **‘Immune epitopes’:** Changed to “accessible epitopes on immune cells”. The selection is for accessible epitopes on immune cells not to be confused by immune epitopes commonly used as epitopes targeted by the immune system. We have shown that we can identify target proteins for which these epitopes belong and that they can be demonstrated to be on the immune cell of interest.
- **‘in the tumor microenvironment’:** This publication specifically addresses the tumor microenvironment. However, we did select for lymph nodes in parallel due to the desire to compare selection. This technology is not limited only to cancer, but it is the primary area of interest for us to further develop the methodology for discovery and drug development.
- **This publication details the methodology and results for simultaneous selection, validates enrichment parameters for parallel immune cell subtypes, shows that some nanobodies may be functionally active, validate that enriched nanobodies can be identified and shows binding to the target cell of interest, and we show how we will use this information for the future development of computational predictions.**

2. Figure 1. No detailed or traceable description/reference/methods were described for construction of the naïve llama phage display library. As this work formed the foundation of the studies, it is essential to be included with adequate details.

This is an excellent point. We recognize that more details would be helpful to a broader audience for Nature Communications. We have added the following description to the methods section...

The llama VHH antibody library was purchased from Abcore Inc. (Ramona, CA). The library was generated using RNA isolated from peripheral blood mononuclear cells (PBMC) cells that were collected from 20 naïve (non-immunized). A combined total of 1.5 x 10⁹ PBMC cells (approximately 1 x 10⁸ per llama) were isolated for RNA production. Total RNA was purified with phenol/chloroform extraction, followed by silica-spin column method. Total RNA was eluted with RNase-free H₂O. Quality of RNA was evaluated by OD260/280 ratio (>1.9) and agarose gel electrophoresis (non-denaturing). RNA concentration was estimated using formula of 1.0 OD260 = 40µg/ml. Library construction was done by QooLabs Inc. (Carlsbad, CA). Reverse transcription and a primer specific for llama IgG was used to prime the total RNA to generate full length cDNA. The quality of cDNA was evaluated by PCR using llama IgG heavy chain specific primers spanning the variable region and the constant region. Products of VH and VHH with expected sizes were amplified from the cDNA using primers to enable cloning into phage

*display vector pADL20. Products of VHH were further purified and modified with *sfiI* sites for cloning into the pADL20c from Antibody Design Labs (ABDL, San Diego, CA) phage display vector. The ligated DNA was then transformed into TGI cells. A total of 2×10^9 independent clones were obtained for the library. Phage were then amplified to generate phage lysates with a titer of 2.5×10^{11} . The VHH phage library was quality controlled by rescue using helper phage VCSM13. One hundred independent clones were selected randomly, and DNA inserts of each clone were sequenced. Over 90% of the clones represent putative immunoglobulin sequences in correct length and reading frame.*

Abcore focuses activities involving llamas and maintains a llama farm while their subcontract with QoolAbs focuses on library construction, so Abcore was unable to provide additional details to us. Therefore, we obtained details from QoolAbs, who shared information on one of the libraries made with 16 llamas. We confirmed with Abcore that our library is their 20 llama library that they had QoolAbs prepare. Below is from the data sheet Abcore provided.

“Llama VHH Single Domain Antibody Library Construction and Screening Services

Abcore has developed proprietary procedures for VHH single domain antibody production. Our optimized library construction and screening processes with phage display technology guarantee successful isolation of high affinity VHH clones in a short time. We have successfully produced multiple clones for a variety of antigens, ranging from large protein molecules (>400kD) to small haptens (small chemical molecules of ~200 dalton), many of the binders have sub-nanomolar affinities measured by ELISA.

Why Abcore?

- 1. We have one of the largest llama immunization facilities in the nation. Scientists and staff at Abcore are very knowledgeable on immunization of a variety of animals. Their experiences are indispensable on generating good VHH antibodies since high antibody titer in the animal blood is the first step to guarantee the success of VHH antibody isolation.*
- 2. We use the freshest cells and best quality RNA to make the library. Since VHH single domain library is constructed with mRNA that encodes the antibody genes, it is critical to capture the expression profile of the antibody producing B cells. It is also well known that expression profile and mRNA composition change rapidly once the cells are taken out of their native environment. Therefore, we make every effort to preserve the health of cells before lysing them for RNA isolation. All animal bleed are processed in the same day (usually within 2-4 hours) to purify the peripheral mononuclear cells (PBMC).*
- 3. We use the best quality RNA to make the cDNA. All RNAs are double-purified with phenol-chloroform extraction and spin columns. RNA qualities are examined by electrophoresis before being used to make cDNA.*
- 4. Our proprietary PCR primers ensure the maximum coverage of VHH repertoires. We have performed extensive bioinformatics research on VHH genes and designed novel primers for RT-PCR to amplify VHH cDNA. VHH libraries constructed with these primers have much higher diversity and larger coverage of the VHH repertoires. In fact, we have identified several pico-molar affinity VHH binders that have unique sequences and would have been missed if published primers were used.*

5. We deliver libraries with large numbers of independent clones. Our proprietary PBMC isolation and library construction protocol routinely yields $>1 \times 10^8$ PBMC cells from each production bleed.
6. We understand client's need. Our principal scientist will discuss the project directly with clients to custom design each project. We offer flexible service schedules to fit client's lab setup and budget (see additional services).
7. Check out our newest VHH library from non-immunized llamas! We have collected over 2.0×10^9 peripheral mononuclear cells from 20 non-immunized llamas. Using our proprietary VHH cloning protocols, a large naïve VHH library with over 2×10^9 independent clones was constructed in phage display vectors. “

Sample Data:

Specificity and affinity determination of three clones of VHH antibodies for AG05 (small molecule, MW ~390)

3. While the authors claimed that the binding of a particular VHH to the single cells can correlate to the RNA expression changes in the TME immune cells, the adequate data was not shown in details in the manuscript. It would be more convincing to thoroughly present the data, if available.

Thank you, these comments are very helpful. Upon reviewing this section, it is understood there were many details that may not have been clearly conveyed in Figure 4. We added additional details, data, and justification to the results section. The results are now displayed across Figure 4-6. A substantial revision has been made to communicate the results of these experiments more effectively. We have edited the prose to not overstate what the experiment is

showing because the following was not the intent of the figure “a particular VHH to the single cells can correlate to the RNA expression changes in the TME immune cells”.

- **Details:** In Fig. 4A we outlined the experiment where we injected BP1 phage library into mice, two hours later harvested and sorted cells, then took enriched pools for CD8 and CD11c cells to injected again and then perform scRNAseq.
- **Data and results:** We have added new data to show that the cell type enriched phage can alter the behavior of target cells.
 - **Figure 4** focuses on the scRNAseq that was done and that there are pharmacodynamic effects on DC and CD8 T cell pathways. In addition, the heat map shows how CD11c and CD8 phage alter transcripts in the target cells of interest compared to other cells.
 - In **Figure 5**, we more deeply evaluate the effects on target DCs of the enriched phage pool selected against CD11c cells. The CD8 and CD11c samples led to increased activation, type 1 interferon, maturation signals, and regulatory genes. These expression profiles are contributed by the expected DC populations (Fig. 5E).
 - In **Figure 6**, we dissect the CD8 T cells of injected mice and reveal greater immune response in samples based upon biologic processes, the active subpopulation contains mostly cells with CD11c or CD8 phage injected. This population is more active with greater immune checkpoints, cytokine, co-stimulation, etc.
- **Justification:** We hope that these revisions show that there is clear immune cell specific impact of the enriched phage pool on the target cells. In many of these comparisons there is enrichment of samples injected with CD11c and CD8 pools compared to the PBS, bacteriophage without nanobody insert, or CD45 enriched phage. These are exciting observations that we are following up for a subsequent study focused on phage/nanobodies that can activate an immune response.

4. It is intriguing to find Nb1 antigen target, PHB2, and impressive to identify the antigen and convincingly demonstrate the binding of Nb1 to PHB2. However, it is not clear whether binding of the immune cells by Nb1 affects the dendritic function in antigen presentation or immune cell activation.

Thank you for the excellent points. It was not our goal to show that Nb1 can alter DC function in antigen presentation or immune cell activation in Figures 5-7 (now 7-9). Rather our goal was to show that we can identify dendritic cell binding nanobodies and identify their target antigens, then recombinantly express the nanobody and verify binding. This is the first stage of the complete pipeline. We agree that having functional nanobodies is high priority and of great interest. This is the primary topic of the ongoing work, where we are working on a publication implementing a robust battery of functional assays to describe downstream interventions of identified nanobody targets. We have added more scRNAseq data in Figure 4-6 from *in vivo* experiments to show the nanobodies can induce transcriptomic changes that may be functionally related. We observed that the data needed to rigorously demonstrate functional

impacts of cell type-specific nanobody binding is voluminous and possibly detract from describing establishment of our methodological pipeline; instead, it will be the focus of a separate, follow-up publication.

5. It is also important to demonstrate whether the INSPIRE method can isolate a VHH that not only targeting an immune cell in TME, but also modulating the function of the immune cell via the binding to improve the anti-cancer immunity. This is a crucial point, because lacking of this does not distinguish the unique benefit of using INSPIRE-seq in identifying the VHH binders to the immune cells in TME in vivo, as the simple VHH phage panning ex vivo using the isolated TME immune cells ex vivo is supposed to do the similar or same thing, substantially reducing the impact of the technology.

We agree about the importance of INSPRE-seq to modulate immune function. See our response to point #4, as this is the primary topic of ongoing work. This is a difficult problem that we aim to solve. However, as we note in the introduction, there is great importance to being able to develop a library of nanobodies that have differential binding properties for different immune cell subsets in the native microenvironment.

We do believe there are several unique benefits of INSPRE-seq in vivo selection compared to ex vivo selection strategies.

- 1. In vivo selection technique ensures that targets are accessible as nanobodies must first reach the target after systemic administration.**
- 2. Epitopes may be specific for cellular activities in the microenvironment even if they do not alter the function or activity. Marking the cells can be powerful and still be exploited for therapeutic purposes.**
- 3. Recreating the microenvironment is not easy for *in vitro* or *ex vivo* selection. Therefore, such efforts to be physiologically relevant to the dynamic interactions of the TME with the host immune system may take considerable optimization and are not guaranteed to translate.**
- 4. Selecting nanobodies that have differential enrichment to immune cells in vitro or ex vivo can be done. However, there is no guarantee or verifiable methodology that would ensure the preservation of epitopes and relative expression of target antigens in this type of model system during the selection.**

We have modified the introduction to clarify the advantages of in vivo over in vitro/ex vivo.

In the introduction we say... “Thus, nanobodies can be enriched for target cells and the target antigen can be identified. Further investigation of identified targets can open new avenues for research or drug development. Such drug development can be simplified where the selected nanobody could be the drug, the targeting moiety, or be modified to develop a drug, such as an antibody-drug conjugate.”

Functional nanobodies would be the “drug”. However, specific nanobodies could serve as the “targeting moiety” or be “modified” by standard industrial methods to make a drug.

In addition, it is powerful to have been able to identify these selective libraries simultaneously for multiple different immune cell subsets. The presence of a library of nanobody candidates

that have differential selectivity combined with binding prediction pipelines as we outline in Figure 7 (now 9) have great power to provide off the shelf nanobodies for various applications.

Reviewer #2 (Remarks to the Author):

In this paper, the authors developed INSPIRE-seq to select immune cell-binding nanobodies that penetrate the tumor microenvironment. It may be a good strategy for drug discovery in the future. However, we still have several concerns on this paper:

1. There's a series of writing mistakes in this manuscript. For example, "to for" in the first line of the abstract. The authors should check over the whole manuscript to improve the readability.

Thank you for pointing this out. We have performed a thorough edit and rewrite. Please see the track changes.

2. The authors described "A camelid based VHH nanobody bacteriophage library was derived from peripheral blood mononuclear cells of sixteen non-immunized llamas". However, the immune background seems not included in the manuscript. It should be detailed that how they get the important materials in such a biopanning pipeline.

We have added additional information about the generation and library in the methods. We hope that these edits are acceptable. Please see the additions below.

*The llama VHH antibody library was purchased from Abcore Inc. (Ramona, CA). The library was generated using RNA isolated from peripheral blood mononuclear cells (PBMC) cells that were collected from 20 naïve (non-immunized). A combined total of 1.5×10^9 PBMC cells (approximately 1×10^8 per llama) were isolated for RNA production. Total RNA was purified with phenol/chloroform extraction, followed by silica-spin column method. Total RNA was eluted with RNase-free H₂O. Quality of RNA was evaluated by OD260/280 ratio (>1.9) and agarose gel electrophoresis (non-denaturing). RNA concentration was estimated using formula of $1.0 \text{ OD260} = 40 \mu\text{g/ml}$. Library construction was done by QooLabs Inc. (Carlsbad, CA). Reverse transcription and a primer specific for llama IgG was used to prime the total RNA to generate full length cDNA. The quality of cDNA was evaluated by PCR using llama IgG heavy chain specific primers spanning the variable region and the constant region. Products of VH and VHH with expected sizes were amplified from the cDNA using primers to enable cloning into phage display vector pADL20. Products of VHH were further purified and modified with *sf*I sites for cloning into the pADL20c from Antibody Design Labs (ABDL, San Diego, CA) phage display vector. The ligated DNA was then transformed into TG1 cells. A total of 2×10^9 independent clones were obtained for the library. Phage were then amplified to generate phage lysates with a titer of 2.5×10^{11} . The VHH phage library was quality controlled by rescue using helper phage VCSM13. One hundred independent clones were selected randomly, and DNA inserts of each clone were sequenced. Over 90% of the clones represent putative immunoglobulin sequences in correct length and reading frame.*

3. Since that the biopanning was performed in two mouse model, the cross-species binding ability should be further confirmed. Could it bind to the homologous protein on the surface of human immune cells?

Thank you for the excellent question. We believe many nanobodies will have cross reactivity, but this is not a given. In the case of PHB2, the protein sequence is identical in humans and mice therefore we anticipated similar binding. We did not need to perform docking experiments as the results would be identical. We then performed immunostaining on human cells and observed similar binding of PHB2 antibody and Nb1 as we observed in mouse cell (Sup. Fig. 9)

The following prose was added to the manuscript...

There is much interest in whether Nbs discovered through INSPIRE-seq could translate to human protein targets, thus having cross-species reactivity. The PHB2 human protein sequence is identical to the murine protein so modeling discussed in the next section would be identical. Therefore, we stained human cells with PHB2 antibody and Nb1 to determine if there was cross reactivity. We observed the same pattern of staining on the cellular membranes and as the mouse cells (Sup. Fig. 9A). Mander's colocalization coefficient verified colocalization (Sup. Fig. 9B).

Supplementary Figure 9

4. The authors declared that “greater DC activation pathway enrichment in the CD8 and CD11c samples”. However, genes in Figure 4I seem to be the markers of DC or T cell subsets, which is not surprising to be found that higher in specific cell types. Our concern is that this conclusion is not invalid.

This is a good question and excellent opportunity for clarification. In figure 4I, the left hand of the figure shows the relative expression and percent of cells expressing DC markers when mice

were injected with the various sample phage pools (CD11c, CD8, empty phage, CD45, and PBS alone). There was stronger expression in the mice injected with the CD11c enriched phage pool. Then the right hand of the figure shows that most of those markers are coming from dendritic cell subtypes.

We have now generated Figure 5 that shows both the breakdown of markers of DC function by the sample mice were injected with CD11c enriched phage (Fig. 5D) as well as showing the genes in the immune cell subtypes (Fig. 5E). This should clarify the two aspects that the previous Figure 4I was trying to convey.

5. The authors identified a nanobody that binds to PHB2 on the surface of cDC1. However, they did not evaluate the downstream effect of this binding process. Could it activate the maturation of cDC1 cells? Could it promote the anti-tumor effect of CD8+ T cells? If not, what is the translational value of this nanobody?

These are excellent questions. We do not know if Nb1 alters cDC1s in a unique way. This specific question is out of the scope of the current work and subject of our ongoing work that aims to identify functionally active nanobodies for different immune cell subsets. We described Nb1 and identified the target to show how the pipeline can lead to the identification of and one day the computational prediction of the binding target. Individual nanobodies could have plenty of translational value beyond the promotion of maturation and promoting anti-tumor effects. As we discussed specific nanobodies...

“Further investigation of identified targets can open new avenues for research or drug development. Such drug development can be simplified where the selected nanobody could be the drug, the targeting moiety, or be modified to develop a drug, such as an antibody-drug conjugate.”

Last sentences of the introduction.

Reviewer #3 (Remarks to the Author):

The manuscript, “Simultaneous selection of nanobodies for immune epitopes in the complex tumor microenvironment” describes a new method called INSPIRE-Seq to select for nanobodies in vivo using next generation sequencing. The paper aims to build upon existing in vivo phage display technologies by sorting immune cells in the tumor microenvironment to identify nanobodies enriched in different subpopulations. This work is interesting and is one of few papers that uses an unbiased approach to identify nanobodies selective for certain immune subpopulations in different organ sites. Publication should be considered, although significant issues must be addressed prior:

1. Py8819 and Py117 are used as models of differing immune response to therapy like radiation in the introduction. Throughout the rest of the study, these two tumors are used for biopanning of nanobodies, but there is no indication/discussion of differences between nanobodies in the figures or text. This should be addressed

Thank you for the question. The differential selectivity between immune responsive and unresponsive is one of the primary topics for our follow up paper. However, we have modified

the manuscript in the following places to address this point to clarify and added a figure showing how the ability to select for differences between biologic models and tissues can be powerful. We are excited by these findings and have added the below figure.

New prose at the end of figure 3.

*“We initially hypothesized that the value of parallel enrichment across tissues and tumor types would enable the identification of a library of nanobodies that distinguish unique features of similar cell types in different environments. This could help distinguish important context specific activity. It could also identify antigens specific for such activities. NGS affords this multifaceted evaluation, so we compared unique and overlapping clones identified in biopanning 3 and 4 in Py117 and Py8119 tumors and dLN’s. We identified multiple shared and unique nanobodies for CD45 cells, CD8 and CD11c cells (**Sup. Fig. 6A-C**). These data suggest biology specific evaluation and deep dive could result in important cellular observations. This is a focus of ongoing work and future reports.”*

New figure:

A Shared and unique VHHs between tumor and LN in Py117 and Py8119

B Shared and unique CD8-VHHs between tumor and LN in Py117 and Py8119

C Shared and unique CD11c-VHHs between tumor and LN in Py117 and Py8119

Supplementary Figure 6

2. In Figure 1A as well as other figures, S nozzle N cartoon are mentioned, but they are not explained.

The S nozzle N is supposed to represent magnetic separation using bead-based separation columns. We have modified the figure to clarify this represents magnetic sorting. Below is how we have addressed this.

C NGS Pipeline to Assess for Parallel Enrichment of Nanobodies

3. Additional evidence is needed to determine if the IgGFc-Nb1 binds to Prohibitin-2. Based on the table in supplementary Figure 6B, most of the abundant proteins are intracellular so unlikely to be the binding target, but DNAJC9 has been shown to interact with PHB2 (Bavelloni A. et al. IUBMB 2015). IgGFc-Nb1 could be binding to DNAJC9, and the confocal microscopy and immunoprecipitation/western blotting demonstrate IgGFc-Nb1 is binding to DNAJC9 interacting with PHB2. Confocal microscopy of IgGFc-Nb1 and DNAJC9 would be sufficient to show that IgGFc-Nb1 does not colocalize with DNAJC9. If it does colocalize with DNAJC9, further experiments using blocking by a DNAJC9-specific antibody and PHB2-specific antibody are needed to show antigen specificity.

This is an excellent point, we are appreciative of the reviewer pointing out the relationship of PHB2 with DNAJC9. It has been suggested that DNAJC9 may be the target of IgGFc-Nb1 based on the critical review article published in IUBMB journal by Bavelloni A et al., 2015 (Bavelloni et al., 2015). The review article mistakenly tabulated the interactions of DNAJC9-PHB2 by citing a paper published by Richter-Dennerlein et al. wrongly (Richter-Dennerlein et al., 2014). The fact is Richter-Dennerlein et al, reported the interactions of DNAJC19-PHB2, not DNAJC9-PHB2. Further, DNAJC9 is not reported to interact with PHB2 as per the literature, rather it interacts with histone proteins by being a component in the nucleosome assembly, thus it is not a surprise it would be carried with immunoprecipitation nonspecifically like histones (<https://www.uniprot.org/uniprotkb/Q8WXX5/entry>) (Hammond et al., 2021).

We performed the MS/MS pull down experiment independently three times. We have added additional details about the protocol to the methods section and have the curated list of identified proteins here in the supplemental. The list of peptides was screened for peptides binding to possible target proteins based upon molecular weight ranging from 30-35 kDa, high abundant proteins filtering at 1.0E+05 peptide abundance, and the elimination of common protein contaminants (histone and associated proteins, actin, and ribosome associated proteins) as commonly done in the literature. In each of the three experiments PHB2 was the most abundant protein. DNAJC9 was only the third most abundant in one replicate shown in the supplemental figure and being 5-fold less abundant than PHB2.

Experiment 1

Protein FDR Confidence	Master	Accession	Description	MW in kDa	Gene Symbol	Abundance
High	Master Protein	O35129	Prohibitin-2 OS=Mus musculus OX=10090 GN=Pphb2 PE=1 SV=1	33.3	Pphb2	1.72E+07
High	Master Protein	P47911	60S ribosomal protein L6 OS=Mus musculus OX=10090 GN=Rpl6 PE=1 SV=3	33.5	Rpl6	8.19E+06
High	Master Protein	Q91WN1	DnaJ homolog subfamily C member 9 OS=Mus musculus OX=10090 GN=Dnajc9 PE=1 SV=2	30	Dnajc9	3.43E+06
High	Master Protein	G3UX26	Outer mitochondrial membrane protein porin 2 OS=Mus musculus OX=10090 GN=Vdac2 PE=1 SV=1 Heterogeneous nuclear ribonucleoprotein K (Fragment) OS=Mus musculus OX=10090 GN=Hnrmk PE=1 SV=1	30.4		2.94E+06
High	Master Protein	H3BKD0	SV=1 60S acidic ribosomal protein P0 OS=Mus musculus OX=10090 GN=Rplp0 PE=1 SV=3	33.2		2.26E+06
High	Master Protein	P14869	60S ribosomal protein L7 OS=Mus musculus OX=10090 GN=Rpl7 PE=1 SV=2	34.2	Rplp0	2.05E+06
High	Master Protein	P14148	Voltage-dependent anion-selective channel protein 1 OS=Mus musculus OX=10090 GN=Vdac1 PE=1 SV=3	31.4	Rpl7	1.55E+06
High	Master Protein	Q60932	Malectin OS=Mus musculus OX=10090 GN=Mlec PE=1 SV=2	32.3	Vdac1	1.26E+06
High	Master Protein	Q6ZQ13	Lactate dehydrogenase OS=Mus musculus OX=10090 GN=Ldha PE=1 SV=1	32.3	Mlec	1.18E+06
High	Master Protein	A0A1B0GSR9	Sideroflexin-3 OS=Mus musculus OX=10090 GN=Sfxn3 PE=1 SV=1	34.6	Ldha	1.14E+06
High	Master Protein	Q3U4F0	H-2 class II histocompatibility antigen, A-U beta chain OS=Mus musculus OX=10090 PE=1 SV=1	30.9	Sfxn3	1.03E+06
High	Master Protein	P06344	H-2 class II histocompatibility antigen, A-U beta chain OS=Mus musculus OX=10090 PE=1 SV=1	30	H2-Ab1	8.71E+05
High	Master Protein	P14206	40S ribosomal protein SA OS=Mus musculus OX=10090 GN=Rpsa PE=1 SV=4	32.8	Rpsa ; LOC100045332 ; LOC100505031 ;	7.21E+05
High	Master Protein	P04441	H-2 class II histocompatibility antigen gamma chain OS=Mus musculus OX=10090 GN=Cd74 PE=1 SV=3	31.5	Cd74	7.08E+05
High	Master Protein	P47962	60S ribosomal protein L5 OS=Mus musculus OX=10090 GN=Rpl5 PE=1 SV=3	34.4	Rpl5	6.13E+05
High	Master Protein	P47753	F-actin-capping protein subunit alpha-1 OS=Mus musculus OX=10090 GN=Capz1 PE=1 SV=4	32.9	Capz1	6.08E+05
High	Master Protein	Q922Q8	Leucine-rich repeat-containing protein 59 OS=Mus musculus OX=10090 GN=Lrrc59 PE=1 SV=1	34.9	Lrrc59	5.74E+05
High	Master Protein	Q922Q4	Pyrolysine-5-carboxylate reductase 2 OS=Mus musculus OX=10090 GN=Pyrc2 PE=1 SV=1	33.6	Pyrc2	4.95E+05
High	Master Protein	Q791V5	Mitochondrial carrier homolog 2 OS=Mus musculus OX=10090 GN=Mch2 PE=1 SV=1	33.5	Mch2	4.04E+05
High	Master Protein	Q8R0B4	TAR DNA-binding protein 43 OS=Mus musculus OX=10090 GN=Tardbp PE=1 SV=1	33.3	Tardbp	3.72E+05
High	Master Protein	Q9DB05	Alpha-soluble NSF attachment protein OS=Mus musculus OX=10090 GN=Napa PE=1 SV=1	33.2	Napa	3.62E+05
High	Master Protein	Q9CRD2	ER membrane protein complex subunit 2 OS=Mus musculus OX=10090 GN=Emc2 PE=1 SV=1	34.9	Emc2	3.59E+05
High	Master Protein	Q8CQ08	ATP synthase subunit gamma OS=Mus musculus OX=10090 GN=Atp5c1 PE=1 SV=1	30.2	Atp5c1	3.57E+05
High	Master Protein	Q80T06	Elongation factor 1-delta OS=Mus musculus OX=10090 GN=Ef1d PE=1 SV=1	31.3	Ef1d	3.45E+05
High	Master Protein	P35550	RNA 2'-O-methyltransferase fibrillarin OS=Mus musculus OX=10090 GN=Fib1 PE=1 SV=2	34.3	LOC100044829 ; Fib1 ; LOC102643269	3.34E+05
High	Master Protein	P51881	ADP/ATP translocase 2 OS=Mus musculus OX=10090 GN=Slc25a5 PE=1 SV=3	32.9	Slc25a5	3.04E+05
High	Master Protein	Q35424	Histocompatibility 2, O region beta locus OS=Mus musculus OX=10090 GN=H2-Ob PE=1 SV=1	30.4	H2-Ob	3.02E+05
High	Master Protein	P14901	Heme oxygenase 1 OS=Mus musculus OX=10090 GN=Hmox1 PE=1 SV=1	32.9	Hmox1	2.73E+05
High	Master Protein	Q9CY64	Biliverdin reductase A OS=Mus musculus OX=10090 GN=Bivr2 PE=1 SV=1	33.5	Bivr2	1.71E+05

Experiment 2		Experiment 3				
Protein FDR	Protein FDR	Protein FDR	Protein FDR			
Confidence	Confidence	Confidence	Confidence			
Master	Master	Master	Master			
Accession	Accession	Accession	Accession			
Description	Description	Description	Description			
MW in kDa	MW in kDa	MW in kDa	MW in kDa			
Gene Symbol	Gene Symbol	Gene Symbol	Gene Symbol			
Abundance	Abundance	Abundance	Abundance			
High	Master Protein	035129	Prohibitin-2 OS=Mus musculus OX=10090 GN=Pphb2 PE=1 SV=1	33.3	Pphb2	5.52E+06
High	Master Protein	Q60932	Voltage-dependent anion-selective channel protein 1 OS=Mus musculus OX=10090 GN=Vdac1 PE=1 SV=3	32.3	Vdac1	7.07E+05
High	Master Protein	H3BKD0	Heterogeneous nuclear ribonucleoprotein K (Fragment) OS=Mus musculus OX=10090 GN=Hnrmk PE=1 SV=1	33.2		5.97E+05
High	Master Protein	P51881	ADP/ATP translocase 2 OS=Mus musculus OX=10090 GN=Slc25a5 PE=1 SV=3	32.9	Slc25a5	5.65E+05
High	Master Protein	P14869	60S acidic ribosomal protein PO OS=Mus musculus OX=10090 GN=Rplp0 PE=1 SV=3	34.2	Rplp0	5.55E+05
High	Master Protein	A0A1B0GSR9	L-lactate dehydrogenase OS=Mus musculus OX=10090 GN=LdhA PE=1 SV=1	34.6		5.47E+05
High	Master Protein	P14206	40S ribosomal protein SA OS=Mus musculus OX=10090 GN=Rpsa PE=1 SV=4	32.8	Rpsa; LOC100045332; LOC100505031;	5.44E+05
High	Master Protein	P14901	Heme oxygenase 1 OS=Mus musculus OX=10090 GN=Hmox1 PE=1 SV=1	32.9	Hmox1	3.83E+05
High	Master Protein	Q922Q8	Leucine-rich repeat-containing protein 59 OS=Mus musculus OX=10090 GN=Lrrc59 PE=1 SV=1	34.9	Lrrc59	3.66E+05
High	Master Protein	Q6ZQ13	Malectin OS=Mus musculus OX=10090 GN=Mlec PE=1 SV=2	32.3	Mlec	1.76E+05
High	Master Protein	P47962	60S ribosomal protein L5 OS=Mus musculus OX=10090 GN=Rpl5 PE=1 SV=3	34.4	Rpl5	1.60E+05
High	Master Protein	Q9DB05	Alpha-soluble NSF attachment protein OS=Mus musculus OX=10090 GN=Napa PE=1 SV=1	33.2	Napa	1.54E+05
High	Master Protein	Q9CRD2	ER membrane protein complex subunit 2 OS=Mus musculus OX=10090 GN=Ermc2 PE=1 SV=1	34.9	Ermc2	1.35E+05
Experiment 3						
High	Master Protein	035129	Prohibitin-2 OS=Mus musculus OX=10090 GN=Pphb2 PE=1 SV=1	33.3	Pphb2	5.52E+06
High	Master Protein	P47911	60S ribosomal protein L6 OS=Mus musculus OX=10090 GN=Rpl6 PE=1 SV=3	33.5	Rpl6	2.13E+06
High	Master Protein	G3UX26	Outer mitochondrial membrane protein porin 2 OS=Mus musculus OX=10090 GN=Vdac2 PE=1 SV=1	30.4		1.49E+06
High	Master Protein	H3BKD0	Heterogeneous nuclear ribonucleoprotein K (Fragment) OS=Mus musculus OX=10090 GN=Hnrmk PE=1 SV=1	33.2		1.30E+06
High	Master Protein	P14869	60S acidic ribosomal protein PO OS=Mus musculus OX=10090 GN=Rplp0 PE=1 SV=3	34.2	Rplp0	9.35E+05
High	Master Protein	Q991R1	Sideroflexin-1 OS=Mus musculus OX=10090 GN=Sfxn1 PE=1 SV=3	35.6	Sfxn1	7.85E+05
High	Master Protein	Q60932	Voltage-dependent anion-selective channel protein 1 OS=Mus musculus OX=10090 GN=Vdac1 PE=1 SV=3	32.3	Vdac1	7.51E+05
High	Master Protein	A0A1B0GSR9	L-lactate dehydrogenase OS=Mus musculus OX=10090 GN=LdhA PE=1 SV=1	34.6		7.48E+05
High	Master Protein	P04441	H-2 class II histocompatibility antigen gamma chain OS=Mus musculus OX=10090 GN=CD74 PE=1 SV=3	31.5	CD74	6.74E+05
High	Master Protein	Q80706	Elongation factor 1-delta OS=Mus musculus OX=10090 GN=Ef1d PE=1 SV=1	31.3	Ef1d	4.08E+05
High	Master Protein	Q8R0B4	TAR DNA-binding protein 43 OS=Mus musculus OX=10090 GN=Tardbp PE=1 SV=1	33.3	Tardbp	3.58E+05
High	Master Protein	P14901	Heme oxygenase 1 OS=Mus musculus OX=10090 GN=Hmox1 PE=1 SV=1	32.9	Hmox1	3.48E+05
High	Master Protein	Q8CQ08	ATP synthase subunit gamma OS=Mus musculus OX=10090 GN=Atp5c1 PE=1 SV=1	30.2	Atp5c1	3.47E+05
High	Master Protein	P14206	40S ribosomal protein SA OS=Mus musculus OX=10090 GN=Rpsa PE=1 SV=4	32.8	Rpsa; LOC100045332; LOC100505031;	3.32E+05
High	Master Protein	Q35424	Histocompatibility 2, O region beta locus OS=Mus musculus OX=10090 GN=H2-Ob PE=1 SV=1	30.4	H2-Ob	3.25E+05
High	Master Protein	Q3U4F0	Sideroflexin-3 OS=Mus musculus OX=10090 GN=Sfxn3 PE=1 SV=1	30.9	Sfxn3	3.12E+05
High	Master Protein	Q922Q8	Leucine-rich repeat-containing protein 59 OS=Mus musculus OX=10090 GN=Lrrc59 PE=1 SV=1	34.9	Lrrc59	3.08E+05
High	Master Protein	P06344	H-2 class II histocompatibility antigen, A-u beta chain OS=Mus musculus OX=10090 PE=1 SV=1	30	H2-Ab1	2.83E+05
High	Master Protein	Q6ZQ13	Malectin OS=Mus musculus OX=10090 GN=Mlec PE=1 SV=2	32.3	Mlec	2.61E+05
High	Master Protein	P35550	rRNA 2'-O-methyltransferase fibrillar OS=Mus musculus OX=10090 GN=Hb; LOC100044829; Hb; LOC102643269	34.3	LOC100044829; Hb; LOC102643269	2.50E+05
High	Master Protein	Q9DB05	Alpha-soluble NSF attachment protein OS=Mus musculus OX=10090 GN=Napa PE=1 SV=1	33.2	Napa	2.35E+05
High	Master Protein	P47753	F-actin-capping protein subunit alpha-1 OS=Mus musculus OX=10090 GN=Capza1 PE=1 SV=4	32.9	Capza1	2.17E+05
High	Master Protein	Q922Q4	Pyroline-5-carboxylate reductase 2 OS=Mus musculus OX=10090 GN=Pycr2 PE=1 SV=1	33.6	Pycr2	1.93E+05

Reviewer #4 (Remarks to the Author):

I was asked to evaluate the proteomics portion of this manuscript. Overall, this experiment appears to be well-designed. However, it is not possible to evaluate this work because the authors did not provide the data and experimental details that are necessary to do so. The authors should upload all LC-MS/MS data files, experimental details, and database search results to a public repository such as MassIVE (at UCSD). This is a standard requirement for publication of proteomics data.

Thank you for the comments. When consulting with our proteomic core we overlooked the value of making the data files available publicly. We have now added additional experimental details and have uploaded the data files to the MassIVE public repository.

Dataset accession number: MSV000092458

URL: <https://massive.ucsd.edu/ProteoSAFe/dataset.jsp?task=5404e07d961648b79b5ffb528136cf05>

Link for reviewers to download the data: <ftp://MSV000092458@massive.ucsd.edu>

Password for reviewers: aguilera2023

References:

Bavelloni, A., Piazzini, M., Raffini, M., Faenza, I., and Blalock, W.L. (2015). Prohibitin 2: At a communications crossroads. *IUBMB Life* 67, 239-254.

Hammond, C.M., Bao, H., Hendriks, I.A., Carraro, M., Garcia-Nieto, A., Liu, Y., Reveron-Gomez, N., Spanos, C., Chen, L., Rappsilber, J., *et al.* (2021). DNAJC9 integrates heat shock molecular chaperones into the histone chaperone network. *Mol Cell* 81, 2533-2548 e2539.

Richter-Dennerlein, R., Korwitz, A., Haag, M., Tatsuta, T., Dargazanli, S., Baker, M., Decker, T., Lamkemeyer, T., Rugarli, E.I., and Langer, T. (2014). DNAJC19, a mitochondrial cochaperone associated with cardiomyopathy, forms a complex with prohibitins to regulate cardiopipin remodeling. *Cell Metab* 20, 158-171.

REVIEWERS' COMMENTS

Reviewer #1 (Remarks to the Author):

The authors addressed almost all the questions I raised and the revised manuscript was much improved. As such I recommend publication in the journal.

Reviewer #2 (Remarks to the Author):

Many thanks to the authors for their prompt response. In general, I am still convinced that this study will have a good impact on the field. The authors addressed most of my suggestions. I have only a few suggestions left.

1.Despite the authors' attempts to rectify spelling errors in the initial manuscript, several problems remain. For instance, Figure 1B still contains inconsistencies like "breast tuomr" and "Magnetic breads separation." I respectfully urge a comprehensive review and rectification of text and figure-related errors across the entirety of the manuscript.

2.The interaction process between the described nanobody and the surface PHB2 of cDC1 cells, along with its implications for cDC1 and CD8+ T cell functions, presents a noteworthy area for exploration. In subsequent studies, authors are encouraged to direct their attention towards this facet and make substantive contributions to the advancement and translational potential of innovative therapeutic agents.

Reviewer #3 (Remarks to the Author):

The INSPIRE-Seq technology applies in vivo methodologies to identify and select for nanobodies that selectively bind to different kinds of immune cells in the tumor microenvironment. This allows for both identification of new nanobody tools, potential therapies, and new immune targets such as PHB2 identified in a subset of dendritic cells. This technology could be useful for disease and tumor-specific interactions with the immune system.

The work supports the conclusions and claims, and any concerns have been addressed in this draft. The analysis, methodology, and flow of the paper is sound, and this methodology could be repeated by others for specific use cases.

Reviewer #4 (Remarks to the Author):

I was asked to review the proteomics portion of this paper, in which the authors performed an immunoprecipitation experiment to identify nanobody targets. The isolated proteins were purified on an acrylamide gel, and specific gel bands were identified by mass spectrometry. Although the IP protocol was fairly complete, the mass spectrometry details were still incomplete. The only details provided were the instrument used, database search software used, and that the analysis was performed at their institution's core facility.

The authors uploaded their data to the public repository as requested, but the submission is incomplete. The .raw data files are there, but they did not upload the database search results or more specific information on the mass spectrometry experiments and how the analysis was performed. This is especially important, since the experimental details in the Methods section is lacking. It's impossible to evaluate the assignment of PHB2 from the mass spec data without having the missing information. The authors are urged to provide the missing information to the MassIVE upload.

There's a typo on line 539: "tittering" should be "titering."

A point-by-point response to the reviewers' comments:

Reviewer #1 (Remarks to the Author):

The authors addressed almost all the questions I raised and the revised manuscript was much improved. As such I recommend publication in the journal.

Respond: We sincerely appreciate your thoughtful review and the positive recommendation for the publication of our manuscript. Your feedback has been invaluable in helping us refine our work, and we are delighted to hear that you found the revised manuscript to be much improved.

Reviewer #2 (Remarks to the Author):

Many thanks to the authors for their prompt response. In general, I am still convinced that this study will have a good impact on the field. The authors addressed most of my suggestions. I have only a few suggestions left.

comment #1: Despite the authors' attempts to rectify spelling errors in the initial manuscript, several problems remain. For instance, Figure 1B still contains inconsistencies like "breast tuomr" and "Magnetic breads separation." I respectfully urge a comprehensive review and rectification of text and figure-related errors across the entirety of the manuscript.

Respond: We would like to thank the reviewer for constructive feedback and for bringing the remaining spelling errors and inconsistencies to our attention. We sincerely apologize for any oversight on our part. We accomplished a thorough proofreading process, involving multiple rounds of review by the authors to make sure we corrected all spelling errors.

comment #2: The interaction process between the described nanobody and the surface PHB2 of cDC1 cells, along with its implications for cDC1 and CD8+ T cell functions, presents a noteworthy area for exploration. In subsequent studies, authors are encouraged to direct their attention towards this facet and make substantive contributions to the advancement and translational potential of innovative therapeutic agents.

Respond: We would like to express our gratitude for your insightful suggestion regarding the interaction process between the described nanobody and the surface PHB2 of cDC1 cells and its potential implications for cDC1 and CD8+ T cell functions. Your perspective is highly valuable, and we appreciate your encouragement to explore this exciting avenue further. We completely agree that investigating this aspect could lead to valuable insights and contribute to the advancement of innovative therapeutic agents. We will certainly consider this suggestion in our future research endeavors and aim to make substantive contributions in this area.

Reviewer #3 (Remarks to the Author):

The INSPIRE-Seq technology applies in vivo methodologies to identify and select for nanobodies that selectively bind to different kinds of immune cells in the tumor microenvironment. This allows for both identification of new nanobody tools, potential therapies, and new immune targets such as PHB2 identified in a subset of dendritic cells. This technology could be useful for disease and tumor-specific

interactions with the immune system.

The work supports the conclusions and claims, and any concerns have been addressed in this draft. The analysis, methodology, and flow of the paper is sound, and this methodology could be repeated by others for specific use cases.

Respond: We would like to express our sincere gratitude for your positive assessment of our manuscript and for your valuable comments. Your feedback is greatly appreciated, and we are pleased to hear that you find our INSPIRE-Seq technology to be a promising approach for identifying and selecting nanobodies targeting immune cells in the tumor microenvironment.

Reviewer #4 (Remarks to the Author):

comment #1 I was asked to review the proteomics portion of this paper, in which the authors performed an immunoprecipitation experiment to identify nanobody targets. The isolated proteins were purified on an acrylamide gel, and specific gel bands were identified by mass spectrometry. Although the IP protocol was fairly complete, the mass spectrometry details were still incomplete. The only details provided were the instrument used, database search software used, and that the analysis was performed at their institution's core facility.

Respond: We would like to thank the reviewer for valuable feedback regarding the proteomics portion of our manuscript. We appreciate your careful evaluation of our work, and your comments are highly constructive. We have thoroughly revised the mass spectrometry methodology to provide more comprehensive and detailed information, addressing your concerns. Specifically, we have included additional details regarding sample preparation, chromatographic conditions, and data analysis parameters. We believe these enhancements will significantly improve the transparency and reproducibility of our experimental procedures.

comment #2: The authors uploaded their data to the public repository as requested, but the submission is incomplete. The raw data files are there, but they did not upload the database search results or more specific information on the mass spectrometry experiments and how the analysis was performed. This is especially important, since the experimental details in the Methods section is lacking. It's impossible to evaluate the assignment of PHB2 from the mass spec data without having the missing information. The authors are urged to provide the missing information to the MassIVE upload.

Response: We would like to thank the reviewer for valuable feedback. We have addressed the reviewer request by uploading the missing file and providing this information in the supplementary files. We believe this will enhance the transparency and completeness of our data.

comment #3: There's a typo on line 539: "tittering" should be "titering."

Response: We appreciate the reviewer's keen eye for detail, and we have made the necessary correction in the manuscript.